# Leveraging Pre-Trained Tacit Model for Efficient Multi-Agent Coordination

## Abstract

Exploration inefficiency caused by large policy spaces is a common challenge in multi-agent reinforcement learning. Although incorporating prior knowledge has been demonstrated to improve exploration efficiency, existing methods typically model it as intrinsic rewards, which may violate potential-based conditions, leading to policy deviation and hindering optimal policy learning. To address this, we propose a novel two-phase multi-agent learning framework, **PTMC** (**P**re-training **T**acit **M**odel for efficient **C**oordination), comprising pre-training and coordinated training phases. In the pre-training phase, PTMC conducts decentralized agent training by integrating general prior knowledge through tacit rewards, while enhancing model scalability by masking opponent information. During the coordinated training phase, coordinated policy is initialized as the pre-trained tacit model, and a tacit constraint term is incorporated into the optimization objective to preserve advantageous tacit behaviors while enabling task-specific adaptation. It is worth emphasizing that the pre-training phase of PTMC is highly efficient, constituting only a minor fraction of the total training time compared to the coordinated training. Experimental results demonstrate that our approach significantly outperforms state-of-the-art baselines in terms of both coordinated performance and exploration efficiency.

## 1 Introduction

Multi-agent reinforcement learning (MARL) has drawn increasing attention for addressing multi-agent coordination tasks (Cacciamani et al., 2021; Yuan et al., 2023). In contrast to single-agent settings, the presence of multiple agents leads to exponential growth of the joint state-action space, thus vastly expanding the policy space (Zhang et al., 2024; Chai et al., 2024). This makes efficient exploration a critical issue for policy optimization in MARL.

A common strategy to mitigate exploration inefficiency is to regularize the learning process by incorporating prior knowledge. For instance, existing studies introduce appropriate handcrafted rewards to supplement the environment reward (Jo et al., 2024; Hou et al., 2025; Li et al., 2024). By forming such a composite reward function, agents are guided toward exploring more meaningful policy spaces, which reduces the search space and improves training efficiency. To preserve the optimal policy when applying reward shaping, a necessary condition is that the shaping reward be expressible as the difference in potential function values between consecutive states (i.e., potential-based reward shaping). Other transformations of the reward function may alter the relative values of state-action pairs and lead to suboptimal policies (Ng et al., 1999; Mannion et al., 2017).

However, in most tasks we possess form of common-sense prior knowledge rather than domain-specific prior knowledge. Such common-sense knowledge typically cannot be directly applied to complete the task but instead serves to facilitate task completion. Consequently, incorporating general prior knowledge—whether common-sense or domain-specific—into training as a potential-based shaped reward remains highly challenging. This raises a critical question: *How can we effectively integrate such general prior knowledge into the learning process while preserving the pursuit of an optimal policy, thereby enhancing exploration efficiency?*

To address the challenge of inefficient exploration in MARL, we introduce **P**re-training **T**acit **M**odel for efficient **C**oordination (**PTMC**), a novel framework that provides a method for incorporating general prior knowledge into the learning process. Inspired by spontaneous tacit coordination in

teamwork (Reber, 1989; Tee & Karney, 2010), we refer to the general prior knowledge as *tacit consensus* among agents and construct the corresponding tacit reward function accordingly. In the pre-training phase, agents are trained in a decentralized manner under the guidance of tacit reward, efficiently yielding a tacit model with minimal computational cost. During the subsequent coordinated training phase, the pre-trained tacit model is used both to initialize the coordinated policy and to incorporate a tacit constraint term into the optimization objective. Collectively, the two-phase framework facilitates efficient policy discovery and promotes stable cooperation among agents.

Our main contributions are three-fold: (1) We formalize the notion of "tacit consensus" and construct a corresponding tacit reward with semantic interpretation and formal definition, enabling the incorporation of general prior knowledge into learning process. (2) We propose a tacit pre-training mechanism for MARL, where single-agent training is guided by the tacit reward to produce tacit behavior, with model scalability enhanced by masking opponents information. (3) Within the centralized coordinated training phase, we integrate a tacit constraint term into the optimization objective, allowing the policy to selectively retain beneficial tacit behaviors.

Empirically, we evaluate PTMC on challenging StarCraft II micromanagement tasks (Samvelyan et al., 2019; Ellis et al., 2024) and Predator–Prey scenarios. PTMC outperforms baseline methods in both coordinated performance and training efficiency. Ablation studies validate the contribution of each component, and visualizations reveal that PTMC exhibits improved exploration efficiency.

## 2 RELATED WORK

In MARL, the expansive joint state-action space leads to a vast policy space, resulting in inefficient exploration. In single-agent reinforcement learning, this issue is often mitigated through warm-starting or regularizing the learning process using prior knowledge (Taiga et al., 2023; Schwarzer et al., 2021; Nair et al., 2020; Ramrakhya et al., 2023; Bruce et al., 2023; Zhou et al., 2023). Recent works extend this idea to MARL by introducing intrinsic motivation to augment environment rewards during training (Zheng et al., 2021; Li et al., 2024; Jeon et al., 2022). For example, FoX (Jo et al., 2024) introduces formation-based rewards to guide exploration towards meaningful states under specific formations. E2M (Hou et al., 2025) employs intrinsic motivation to encourage exploration while avoiding overly conservative policies. However, such handcrafted rewards may violate potential-based reward conditions, undermining convergence guarantees and impeding optimal policy learning (Ng et al., 1999; Mannion et al., 2017). To this end, we propose a pre-training mechanism that introduces general prior knowledge to avoid convergence risks.

Existing pre-training approaches in MARL primarily focus on acquiring shared knowledge to facilitate online fine-tuning across multiple downstream tasks (Meng et al., 2023b; Wang et al., 2025). For example, M3 (Meng et al., 2023a) learns transferable high-level policy representations and integrates them into subsequent training, while recent work extends this idea by leveraging diverse reward-level data to pre-train policies with broader applicability (Meng et al., 2024). However, these paradigms typically depend on similar data sources and environment rewards, limiting the incorporation of diverse information such as common-sense prior knowledge. To address this, we introduce tacit reward functions to encode prior knowledge for decentralized pre-training, followed by centralized coordinated training to learn efficient policies. In contrast to conventional pre-training methods that rely on large-scale data (Baker et al., 2022; Pertsch et al., 2021; Fan et al., 2022), our approach focuses on simpler single-agent tasks and requires significantly fewer training steps.

## 3 PRELIMINARIES

### 3.1 PROBLEM FORMULATION

The multi-agent coordinated task can be formalized as a Decentralized Partially Observable Markov Decision Process (Dec-POMDP) (Oliehoek et al., 2016), defined by the tuple $G = \langle N, \mathbb{S}, \mathbb{O}, \mathbb{A}, P, \mathbb{R}, \gamma \rangle$. Here, $N$ denotes the set of agents with $n = |N|$, $\mathbb{S}$ is the global state space. $\mathbb{O} = \{o_i\}_{i=1}^{n}$ is the joint observation space, where $o_i$ is the local observation of agent $i$. The joint action space is $\mathbb{A} = \{a_i\}_{i=1}^{n}$, comprising individual actions of each agent. In most multi-agent benchmarks, agents share a common environment reward $r^t = \mathbb{R}(s^t, \mathbf{a}^t)$. The environment reward is determined by the transition function $P(s^{t+1}|s^t, \mathbf{a}^t)$ based on the change in global state and joint

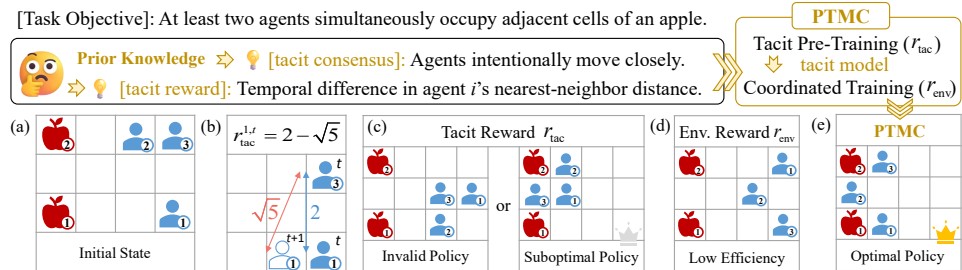

Figure 1: A toy example illustrating of tacit consensus formation and tacit reward design in multi-agent coordination tasks. (a) Initial positions of agents and apples, and the environment reward is provided for the successful capture of an apple. (b) Using agent 1 as an example, its tacit reward $r_{\text{tac}}^{1,t}$ is computed as the temporal difference in distance between agent 1 and agent 3. (c) Training solely with tacit reward may yield invalid or suboptimal policies. (d) Training solely with environment reward results in inefficient policy learning. (e) By leveraging the tacit model, PTMC integrates both rewards and accelerates convergence toward the optimal policy.

action $\mathbf{a}^t$. The discount factor $\gamma \in [0, 1)$ determines the weight of future rewards. In MARL, each agent learns a policy $\pi_\theta(a_i|o_i)$ that maps its local observation $o_i$ to an action $a_i$, aiming to maximize the expected cumulative discounted reward:

$$J(\theta) = \mathbb{E}_{\pi_\theta} \left[ \sum_{t=0}^{\infty} \gamma^t \cdot R(s^t, \mathbf{a}^t) \right]. \tag{1}$$

### 3.2 KEY CONCEPTS AND DEFINITIONS

In multi-agent coordinated tasks, agents can improve efficiency by leveraging shared prior knowledge toward collective objectives. We term the shared understanding of prior knowledge among multi agents as *tacit consensus*. Building on this concept, we construct the corresponding *tacit reward* function and provide a method for deriving tacit rewards from tacit consensus. Specifically, tacit consensus typically entails an *advantageous configuration* $C_{\text{adv}}$, representing a specific agent formation that yields a cooperative advantage. Depending on tasks, the *configuration* $C$ can manifest as either a particular spatial arrangement, a temporal sequence of actions among agents, or both. During training, the configuration of agent $i$ at time $t$ is denoted by $C_i^t$. The distance between $C_i^t$ and $C_{\text{adv}}$ is quantified as configuration distance. The tacit reward $r_{\text{tac}}$ for each agent is then constructed from the temporal change in configuration distance, quantifying the extent to which agent's behavior converges toward $C_{\text{adv}}$. Formally, the tacit reward is defined as:

$$r_{\text{tac}}^{i,t} = \left\| C_i^t - C_{\text{adv}} \right\| - \left\| C_i^{t+1} - C_{\text{adv}} \right\|, \tag{2}$$

where $\|\cdot\|$ represents a general configuration distance metric.

The behaviors learned under the guidance of this tacit reward are defined as *tacit behaviors*. Although agents make decisions independently, their tacit behaviors collectively foster coordinated team behavior—precisely the outcome that tacit consensus is intended to achieve. Importantly, tacit consensus can be derived from either domain-specific prior knowledge or common-sense priors. Moreover, the approach we describe is not the only method for deriving tacit rewards from tacit consensus; alternative methods can also be integrated into the overall framework.

### 3.3 A TOY EXAMPLE

To further clarify the concept of tacit consensus and illustrate how it is formed, we provide a concrete example, as shown in Figure 1. In this task, the evident prior knowledge is that agents can accomplish the objective more efficiently by moving closer to each other. Building on this insight, we derive the tacit consensus and corresponding tacit reward for this task, as illustrated in Figure 1. Moreover, tacit rewards can also be constructed by defining the task's advantageous configuration. Accordingly, we define "spatial configuration where agents are clustered" as advantageous configuration $C_{\text{adv}}$. At time $t$, the distance between agent $i$ and its nearest teammate is defined as $C_i^t$. Consequently, the tacit reward $r_{\text{tac}}^{i,t}$ is formulated as the temporal reduction in the distance to $C_{\text{adv}}$.

Figure 2: The overall PTMC framework. The colored arrows and blocks depict processes and modules unique to PTMC.

It is important to emphasize that the objective of tacit consensus is not fully aligned with original task objective. For example, reducing pairwise distances among agents is a necessary but insufficient condition for task completion. Relying solely on tacit rewards may cause deviation from the task objective, leading to invalid or suboptimal policy (Figure 1(c)), whereas relying only on environment rewards causes inefficient exploration (Figure 1(d)). Our method combines both, using tacit reward as auxiliary guidance to accelerate convergence toward the optimal policy (Figure 1(e)).

## 4 METHOD

This section presents the proposed method, PTMC, a learning framework that enhances coordination efficiency in multi-agent settings by leveraging prior knowledge. As shown in Figure 2, PTMC comprises two training phases. In the decentralized tacit pre-training phase, agents are trained individually while masking opponent-related information, yielding tacit behaviors that generalize across diverse coordinated scenarios. In the centralized coordinated training phase, we introduce a tacit constraint term into the optimization objective, defined as the product of a binary gating function and a deviation regularization term. This constraint is incorporated to selectively leverage the beneficial coordination encoded in the pre-trained tacit model, promoting efficient learning of high-return coordinated policies.

### 4.1 TACIT PRE-TRAINING

Under the decentralized training paradigm, the tacit reward guides agent toward efficient tacit behaviors, facilitating the integration of prior knowledge. A simple yet effective masking strategy is employed to train the tacit policy independently of specific opponent settings. Additionally, a "tacit metric" is introduced to quantify the degree of tacit behavior acquisition, enabling adaptive termination of the pre-training phase.

#### 4.1.1 MASK OPPONENTS INFORMATION.

To facilitate the learning of tacit behavior among agents and ensure that the resulting tacit policy is opponent-agnostic, we define the policy of agent $i$ during the tacit pre-training phase as:

$$\pi_i^{\text{tac}} \triangleq \pi_i^{\text{tac}}(a_i \mid f_{\text{mask}}(o_i), h_{\text{mask}}(\widetilde{\mathcal{A}}_i); \theta_{\text{tac}}). \tag{3}$$

Here, the opponent-related components are masked in agent $i$'s observation space $o_i$ and executable action space $\widetilde{\mathcal{A}}_i$. The masking functions $f_{\text{mask}}$ and $h_{\text{mask}}$ are given by:

$$f_{\text{mask}}(o_i) = f_{\text{mask}}([o_i^{\text{ag}}, o_i^{\text{op}}]) = [o_i^{\text{ag}}, 0], \tag{4}$$

$$h_{\text{mask}}\left(\widetilde{\mathcal{A}}_i\right) = h_{\text{mask}}\left(\left[\widetilde{\mathcal{A}}_i^{\text{ag}}, \widetilde{\mathcal{A}}_i^{\text{op}}\right]\right) = \left[\widetilde{\mathcal{A}}_i^{\text{ag}}, 0\right], \tag{5}$$

where $o_i$ is partitioned into two subsets: $o_i = o_i^{\text{ag}} \cup o_i^{\text{op}}$. $o_i^{\text{ag}}$ denotes agent $i$'s own state and its local observations of allied agents, and $o_i^{\text{op}}$ contains local observations of opponents. Similarly, $\widetilde{\mathcal{A}}_i$ is divided as $\widetilde{\mathcal{A}}_i = \widetilde{\mathcal{A}}_i^{\text{ag}} \cup \widetilde{\mathcal{A}}_i^{\text{op}}$. $\widetilde{\mathcal{A}}_i^{\text{ag}}$ represents executable actions related to the agent $i$ and its allies (e.g., movement), and $\widetilde{\mathcal{A}}_i^{\text{op}}$ includes executable actions that interact with opponents (e.g., attack).

By masking opponent-related components during pre-training, the resulting tacit policy acquires opponent-agnostic coordination skills, improving adaptability during subsequent coordinated training. For instance, in the task presented in "A Toy Example" section, the tacit behavior learned during pre-training remains effective despite variations in the number or positions of apples.

### 4.1.2 TACIT PRE-TRAINING OBJECTIVE.

During the tacit pre-training phase, each agent's tacit policy model is trained under the decentralized training paradigm (e.g., IPPO or IQL), where agents rely only on local observations and thus struggle to learn global coordinated behaviors. To address this, we design a tacit reward function $r_{\text{tac}}$ to incorporate global state information, allow agents to access information beyond their observation ranges, and thereby promote effective tacit coordination. Specifically, $r_{\text{tac}}^{i,t}\left(s^t, s_i^{t+1}\right)$ is defined using a counterfactual global state $s_i^{t+1}$, where agent $i$'s state is updated while all allied agents' state remain as in $s^t$. This design ensures that the reward captures only the impact of agent $i$'s action, isolated from the influence of other agents.

For environments with homogeneous teammates, we adopt parameter sharing and optimize a single policy parameterized by $\theta_{\text{tac}}$. In tacit pre-training, the optimization objective is to maximize:

$$J(\theta_{\text{tac}}) = \mathbb{E}_{\pi_{\text{tac}}}\left[\frac{1}{n}\sum_{t=0}^{\infty}\sum_{i=1}^{n}\gamma^t r_{\text{tac}}^{i,t}\left(s^t, s_i^{t+1}\right)\right]. \tag{6}$$

Although both phases employ multi-agent algorithms, tacit pre-training is essentially single-agent learning, without opponent-induced variability, whereas coordinated training involves multi-agent interactions. Due to these factors, the tacit pre-training phase exhibits lower learning complexity, requiring fewer training steps than coordinated training.

### 4.1.3 TERMINATION CRITERIA.

To quantitatively evaluate the real-time effectiveness of tacit pre-training and determine its termination point, we introduce the *tacit metric* $M_{\text{tac}}$. In each training episode, a batch of test trajectories is sampled, and $M_{\text{tac}}$ is computed by accumulating over the samples in the batch as follows:

$$M_{\text{tac}} = \frac{1}{N}\sum_{t=0}^{N-1} H\left(r_{\text{tac}}^t\right), \tag{7}$$

where $H(\cdot)$ denotes the Heaviside step function, $r_{\text{tac}}^t$ is the tacit reward obtained under policy $\pi_{\text{tac}}^t$, and $N$ represents the total number of samples in the test batch.

Throughout tacit pre-training phase, $M_{\text{tac}}$ is monitored to evaluate the degree of tacit behavior acquisition. Once it exceeds a predefined threshold $M_{\text{tac}}^*$, the tacit pre-training process is terminated, indicating that agents have acquired the coordination capability required for tacit behaviors.

## 4.2 COORDINATED TRAINING

During the centralized coordinated phase, the pre-trained tacit policy initializes coordinated policy learning, enabling agents to develop coordination skills efficiently. To facilitate this process, we incorporate a tacit constraint term into the optimization objective, guiding policy updates to selectively preserve beneficial tacit behaviors. Within this module, a binary gating function activates the deviation regularization term only when harmful drift from tacit coordination is detected, ensuring that corrections are applied adaptively while maintaining useful tacit behaviors.

### 4.2.1 COORDINATED TRAINING PROCESS.

During the coordinated training phase, agent $i$'s policy is denoted as $\pi_i^{\text{coor}}$ and initialized from the pre-trained tacit policy $\pi_i^{\text{tac}}$ by setting $\theta_{\text{coor}} \leftarrow \theta_{\text{tac}}$. Unlike the critic used in tacit pre-training, which is trained on individual observations, the critic network in coordinated training is based on the global state. Coordinated training follows the Centralized Training with Decentralized Execution (CTDE) paradigm and is guided by the environment reward $r_{\text{env}}$, computed from joint actions.

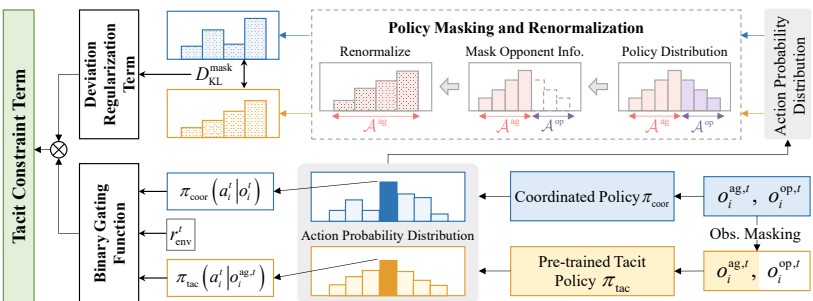

Figure 3: Detailed structure of the tacit constraint term, expressed as the product of a binary gating function and a deviation regularization term.

Although initialization with the pre-trained model incorporates tacit coordination potential, iterative policy updates may compromise the retention of its benefits. To mitigate this, we introduce a tacit regularization term $\mathcal{L}_{\text{tac}}$ (structure shown in Figure 3) that dynamically constrains deviations between $\pi_{\text{coor}}$ and $\pi_{\text{tac}}$ during training. The coordinated objective is formulated as follows:

$$J(\theta_{\text{coor}}) = \mathbb{E}_{\pi_{\text{coor}}} \left[ \sum_{t=0}^{\infty} \gamma^t \cdot r_{\text{env}}^t - \alpha_{\text{tac}} \cdot \mathcal{L}_{\text{tac}} \right], \tag{8}$$

$$\mathcal{L}_{\text{tac}} = G\left(\rho_t, \varepsilon, r_{\text{env}}^t\right) \cdot D_{\text{KL}}^{\text{mask}}\left(\pi_{\text{tac}} \| \pi_{\text{coor}}\right), \tag{9}$$

where $\alpha_{\text{tac}}$ is a hyperparameter that balances the return and the tacit constraint term, and $G(\cdot)$ is a binary gating function (taking values 0 or 1).

### 4.2.2 BINARY GATING FUNCTION.

To enable selective activation and adaptive adjustment during training, we introduce a binary gating function $G(\cdot)$. This function takes as input the deviation term $\rho_t$ between the action probability distributions of $\pi_{\text{coor}}$ and $\pi_{\text{tac}}$, a threshold parameter $\varepsilon$, and the environment reward $r_{\text{env}}^t$. The definition of $G(\cdot)$ is given as:

$$G(\rho_t, \varepsilon, r_{\text{env}}^t) = \begin{cases} 0, & r_{\text{env}}^t \geq 0, \\ 0, & r_{\text{env}}^t < 0 \text{ and } |\rho_t - 1| \leq \varepsilon, \\ 1, & r_{\text{env}}^t < 0 \text{ and } |\rho_t - 1| > \varepsilon, \end{cases} \quad \rho_t = \frac{\pi_i^{\text{coor}}(a_i^t \mid o_i^t)}{\pi_i^{\text{tac}}(a_i^t \mid f_{\text{mask}}(o_i^t))}. \tag{10}$$

A positive environment reward $r_{\text{env}}^t$ indicates effective agent behavior, negating the need for alignment with the pre-trained tacit policy. In contrast, a negative reward reflects suboptimal actions, and we evaluate whether the suboptimal performance results from the current policy deviating from pre-trained tacit policy, by measuring the deviation of $\rho_t$ from 1.

### 4.2.3 DEVIATION REGULARIZATION TERM.

The deviation regularization term $D_{\text{KL}}^{\text{mask}}(\cdot)$ quantifies the divergence between current policy $\pi_{\text{coor}}$ and pre-trained tacit policy $\pi_{\text{tac}}$. During coordinated policy learning, it guides the agent toward actions that promote task completion while also encouraging behavioral alignment with the pre-trained tacit policy. It is formally defined as:

$$D_{\text{KL}}^{\text{mask}} \triangleq D_{\text{KL}}\left(g_{\text{mask}}\left(\pi_{\text{tac}}(\cdot \mid f_{\text{mask}}(o_i^t))\right) \| g_{\text{mask}}\left(\pi_{\text{coor}}(\cdot \mid o_i^t)\right)\right), \tag{11}$$

where $f_{\text{mask}}(\cdot)$ masks opponent-related information as defined in Eq. (4).

Moreover, for the action probability distributions under $\pi_{\text{tac}}$ and $\pi_{\text{coor}}$, actions involving opponent interactions are excluded, ensuring fair alignment since $\pi_{\text{tac}}$ is not trained on such actions. The masking and renormalization function for policy, denoted as $g_{\text{mask}}(\cdot)$, is given by:

$$g_{\text{mask}}\left(\pi(o_i^t)\right) = \frac{m_i[a_i] \odot \pi(a_i \mid o_i^t)}{\sum_{a_i \in \mathcal{A}_i^{\text{ag}}} \pi(a_i \mid o_i^t)}, \quad m_i[a_i] = \begin{cases} 1, & a_i \in \mathcal{A}_i^{\text{ag}}, \\ 0, & a_i \in \mathcal{A}_i^{\text{op}}, \end{cases} \tag{12}$$

where $\odot$ denotes element-wise multiplication, $\mathcal{A}_i$ denotes the full action space, distinct from the executable subset $\widetilde{\mathcal{A}}_i$ used in pre-training. The action space $\mathcal{A}_i$ is divided into: $\mathcal{A}_i = \mathcal{A}_i^{\text{ag}} \cup \mathcal{A}_i^{\text{op}}$.

## 5 EXPERIMENTS

We evaluate PTMC on the StarCraft Multi-Agent Challenge (SMAC) (Samvelyan et al., 2019), SMACv2 (Ellis et al., 2024) and the Predator-Prey environment (Lowe et al., 2017), comparing it against five advanced MARL algorithms. The results demonstrate that PTMC achieves improved learning efficiency, enhanced coordinated performance and scalability. We further conduct ablation study to validate the contributions of key components within PTMC. Additionally, visualizations are provided to illustrate the advantage of PTMC in high-return state exploration.

### 5.1 COMPARATIVE EVALUATION

#### 5.1.1 BASELINE.

We compare PTMC against five well-established baselines, encompassing both value-based and policy-gradient methods. Several of these baselines are specifically designed to enhance exploration efficiency from different perspectives:

- **AIR** (Zhou et al., 2025) improves exploration through identity recognition and adaptive modulation of exploration mode and intensity.
- **GoMARL** (Zang et al., 2024) promotes exploration via automatic agent grouping.
- **MAT** (Wen et al., 2022) reformulates joint policy optimization as a sequential advantage-guided process to enhance exploration and convergence.
- **QMIX** (Rashid et al., 2018) factorizes the joint action-value via a mixing network under CTDE, enabling decentralized greedy policies trained with a global reward.
- **MAPPO** (Yu et al., 2022) applies PPO in a CTDE setting with a centralized critic and decentralized actors, providing a strong and stable baseline.

All algorithms are open-source, with finetuned hyperparameters for optimal performance. Among them, AIR and GoMARL are QMIX-based variants, while MAT is based on MAPPO.

#### 5.1.2 ENVIRONMENT.

We evaluate our approach on SMAC, SMACv2 and Predator-Prey environments. SMAC and SMACv2 are cooperative multi-agent benchmarks, where two opposing teams engage in combat: one controlled by built-in game bots and the other by MARL algorithms. *Notably, we randomize the initial positions of agents in SMAC maps to increase the difficulty of the scenarios.* Predator-Prey focuses on coordination, requiring agents (predators) to capture stags and hares in a 25×25 grid. Stags must be captured cooperatively by two agents (reward: 10), whereas hares can be captured by a single agent (reward: 2). All environments employ global rewards to reflect overall system performance. Appendix B.1 and B.2 details the environment-specific definitions of the tacit consensus and the corresponding tacit reward employed in PTMC. Appendix C provides a detailed description of the environment settings. Appendix D presents the results of tacit pre-training, along with the scenario-specific settings for the predefined threshold of the tacit metric.

#### 5.1.3 PERFORMANCE.

Since our method is compatible with both QMIX-based and MAPPO-based algorithms, we implement PTMC on each framework and conduct comparative evaluations. Figure 4 presents the learning curves for three SMAC and three SMACv2 scenarios. Although MAPPO-based and QMIX-based algorithms exhibit distinct performance in SMACv2, both PTMC-MAPPO and PTMC-QMIX consistently outperform their respective baselines. While MAPPO and QMIX show stable learning, they exhibit lower efficiency and suboptimal final performance. AIR displays training instability, as reflected by large confidence intervals. MAT shows limited effectiveness across all scenarios. Although GoMARL performs competitively in SMACv2, it demonstrates limited learning efficiency and low win rates in SMAC. In contrast, PTMC-MAPPO and PTMC-QMIX achieve higher learning efficiency, improved final win rates, and enhanced training stability across all tasks.

Figure 5 shows the total training steps to reach equivalent mean return in four Predator–Prey tasks. Notably, PTMC employs the same pre-trained tacit model across all experiments to facilitate coordination among ten agents. Across scenarios varying in prey numbers (more or fewer than predators)

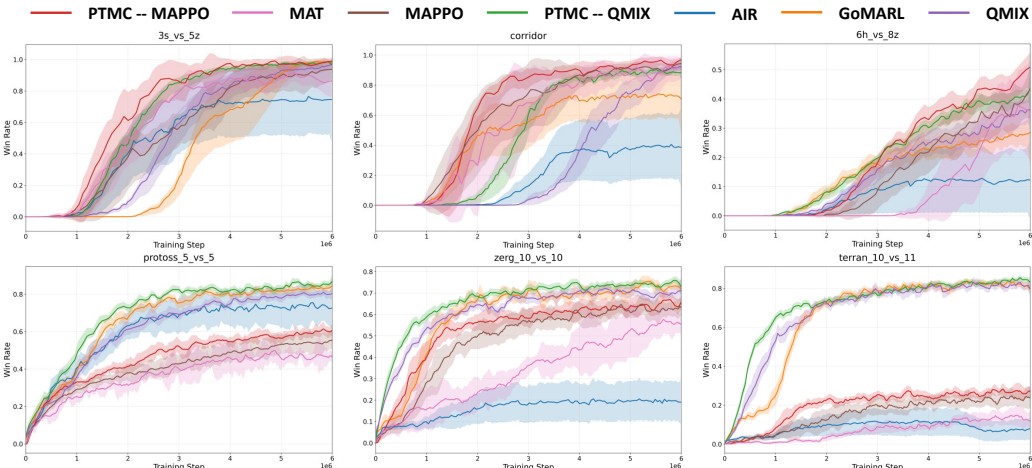

Figure 4: Comparison of training performance in SMAC and SMACv2 over 6M steps. The top row presents results for three SMAC scenarios, and the bottom row for three SMACv2 scenarios. Solid curves indicate the mean across five random seeds and shaded regions denote confidence intervals.

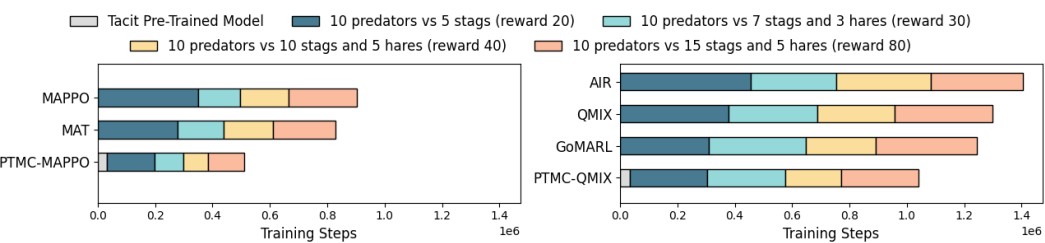

Figure 5: Comparison of training efficiency in Predator–Prey tasks. The bars represent the total number of training steps required to achieve a specified mean return across four scenarios, with PTMC results including its tacit pre-training phase.

and the composition of stags and hares, PTMC-MAPPO and PTMC-QMIX both achieve faster convergence and superior performance compared to their baselines. These results demonstrate the scalability of the tacit pre-training and the effectiveness of PTMC. Additional results for comparative evaluation are provided in Appendix E.

## 5.2 ABLATION STUDIES

We conduct ablation study on 3s_vs_5z to evaluate the contributions of key components in PTMC, where PTMC is built on MAPPO. Specifically, "PTMC w/o Constr." removes tacit constraint term; "PTMC w/o Pretr." omits actor network initialization from tacit pre-training ; and "PTMC w/o BinGate." removes binary gating function to assess the impact of selective constraint enforcement. As shown in Figure 6, PTMC consistently outperforms all ablated variants. Notably, "PTMC w/o Pretr." performs significantly worse, underscoring the importance of pre-trained initialization. PTMC exhibits an early-stage advantage over "PTMC w/o Constr.", validating the ben-

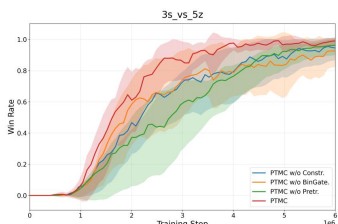

Figure 6: Ablation study of PTMC on the 3s_vs_5z map.

efit of introducing tacit constraint term. Additionally, the performance drop of "PTMC w/o Bin-Gate." suggests that indiscriminate constraint may impair learning due to inaccurate loss estimation. Extensive ablation studies on additional scenarios are presented in Appendix F. In addition, Appendix G reports the evaluation of the parameter setting for $\alpha_{\text{tac}}$.

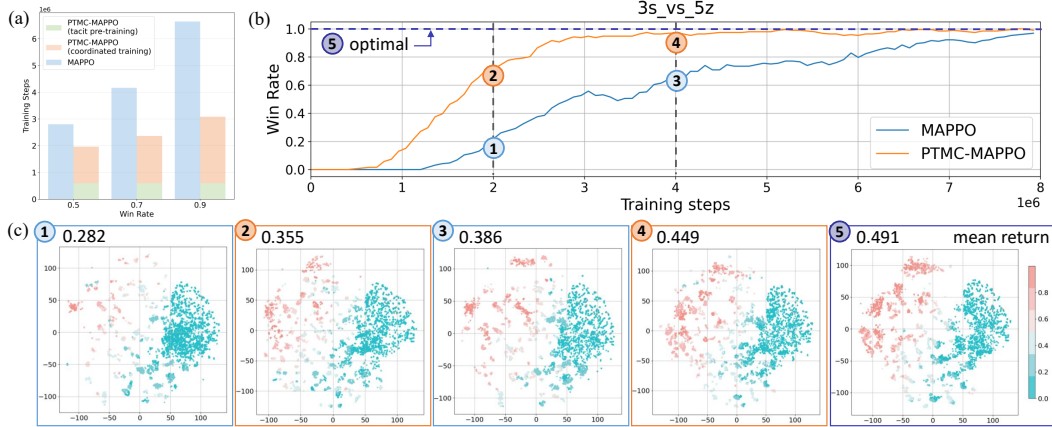

Figure 7: Visualization of exploration efficiency on the `3s_vs_5z` map. (a) Training steps required by PTMC and MAPPO to achieve the same win rate. (b–c) Each of the five subplots in (c) corresponds to a marked point in (b). Within each subplot, scatter points represent the 2D t-SNE embeddings of states, color-coded by the normalized mean return on a gradient from light blue (low return) to pink (high return), with the average value indicated in the top-left corner.

## 5.3 VISUALIZATION ANALYSIS

We evaluate PTMC and MAPPO on `3s_vs_5z` map in SMAC under identical training configurations, where PTMC is built on the MAPPO framework. As shown in Figure 7(a), the total training steps of PTMC across both phases remain fewer than those required by MAPPO, with the difference increasing as the target win rate rises, indicating the superior training efficiency of PTMC. To further investigate the cause of this efficiency gap, we analyze the efficiency of exploring high-return states as training progresses. Specifically, we collect all global states encountered during 32 evaluation episodes for both methods, using the same random seed. Each state is embedded into a low-dimensional space via t-SNE. To ensure consistent relative positioning, we jointly embed five representative state groups into a shared t-SNE space: MAPPO and PTMC at 2M and 4M steps, and PTMC at 10M steps (approximately optimal model).

In Figure 7(c), the state distributions increasingly shift toward high-return regions as training progresses, with Subplot 5 (optimal) showing the highest concentration of high-return states. At 2M steps (Subplots 1 and 2), PTMC already covers a wider range of high-return states, while MAPPO remains concentrated in low-return areas. Moreover, PTMC achieves a significantly higher average normalized mean return than MAPPO at 2M steps, confirming the effectiveness of the tacit pre-training mechanism in guiding exploration toward high-return states. By 4M steps (Subplots 3 and 4), PTMC's state distribution closely aligns with the optimal model and clearly outperforms MAPPO in high-return coverage. Additional visualizations and state coverage comparisons with optimal model are provided in Appendix H.

## 6 CONCLUSION

In this paper, we propose PTMC to improve inter-agent coordination and facilitate efficient policy discovery, thereby enhancing exploration efficiency in MARL. PTMC adopts a two-phase paradigm comprising tacit pre-training and coordinated training, where prior knowledge is encoded as a tacit reward. The tacit reward guides decentralized pre-training to learn individual tacit policies. Building on the pre-trained tacit model, PTMC incorporates a tacit constraint term into the optimization objective, enabling the policy to selectively retain beneficial tacit behaviors. Experiments show that PTMC achieves superior learning efficiency, improved coordinated performance, and scalability across diverse tasks. Ablation studies and visualizations further validate the contributions of key components and the overall effectiveness of PTMC in guiding exploration.

## 7 ETHICS STATEMENT

We acknowledge that all authors of this work have read and commit to adhering to the ICLR Code of Ethics. We explicitly confirm our compliance with the Code throughout the submission, review, and discussion processes.

## 8 REPRODUCIBILITY STATEMENT

We have made every effort to ensure the reproducibility of our results. The source code is provided in the "Supplementary Material", together with the modified SMAC map files (.SC2Map) used in our experiments. The detailed settings of the threshold parameter $M_{\text{tac}}^*$ in the pre-training phase for both SMAC and predator-prey tasks are described in Appendix D. The design of the key parameter $\alpha_{\text{tac}}$ in the coordinated training phase, as well as its performance under different values, is thoroughly analyzed in Appendix G.

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

## APPENDIX CONTENTS

# A  LLM USAGE STATEMENT

We employed large language models (LLMs) solely for translation and language polishing of the manuscript. No part of the research design, problem formulation, experimental setup, or result interpretation was generated or influenced by LLMs. All scientific contributions, ideas, and analyses are entirely the authors' own.

# B  DEFINITION OF TACIT REWARDS BASED ON DIFFERENT ENVIRONMENTS

## B.1  SMAC

Our intuitive insight is that agents can achieve the overall objective more efficiently by forming advantageous spatial relationships, specifically, by enabling the multi-agent system to locally aggregate more agents than its opponents. Considering the partial observability in SMAC environments, we define the *tacit consensus* among agents as maintaining mutual observability while preserving a stable inter-agent distance over time. Due to the complexity of the SMAC environment, we categorize four typical cases and define their corresponding *advantageous configuration* $C_{\mathrm{adv}}$ and *agent-specific configuration* $C_i^t$, where agent $i$ denotes a representative agent from multiple agents. Additionally, we introduce a parameter term $\lambda^{i,t}$ before the tacit reward to dynamically adjust its weight during training, thereby improving learning efficiency.

**Case (a):** In this case, the distance between every pair of agents in the global state exceeds their respective perception ranges, resulting in a lack of mutual observability. To establish the aforementioned tacit consensus, agents should individually converge toward a fixed point to rapidly enter each other's observation range. Accordingly, the *advantageous configuration* $C_{\mathrm{adv}}$ is defined as the minimum distance between agent $i$ and the fixed point, while the *agent-specific configuration* $C_i^t$ denotes the distance between agent $i$'s current position and the fixed point. Formally, $C_{\mathrm{adv}}$ and $C_i^t$ are defined as:

$$\begin{cases} C_i^t = \left\| \varphi_i^t - \varphi_{\mathrm{pt}} \right\|_2 \\ C_i^{t+1} = \left\| \varphi_i^{t+1} - \varphi_{\mathrm{pt}} \right\|_2 \\ C_{\mathrm{adv}} = \left\| \varphi_{\mathrm{cls}} - \varphi_{\mathrm{pt}} \right\|_2, \end{cases} \tag{13}$$

where the position of agent $i$ is denoted by $\varphi_i(o_i, s)$, which is abbreviated as $\varphi_i$ for simplicity in the aforementioned definitions. $\varphi_{\mathrm{pt}}$ denotes the position of the fixed point, $\varphi_{\mathrm{cls}}$ refers to the position closest to the fixed point.

The parameter term $\lambda^{i,t}$ of case (a) is defined in 14. The tacit reward term $r_{\mathrm{tac}}^{i,t}$ is defined in 15.

$$\lambda_a^{i,t} = \min \left\{ \frac{\min\limits_{k \notin \rho_i^t} \left\| \varphi_i^t - \varphi_k^t \right\|_2 - d_i}{\beta d_i - d_i}, 1 \right\}, \tag{14}$$

$$r_{\mathrm{tac}}^{i,t} = \lambda_a^{i,t} \times \left( \left\| C_i^t - C_{\mathrm{adv}} \right\| - \left\| C_i^{t+1} - C_{\mathrm{adv}} \right\| \right), \tag{15}$$

where the set $\rho_i$ denotes the collection of agents within the perceptual range of the agent $i$. The parameter $\beta > 1$ is introduced to define the upper bound of $\lambda$. The term $d_i$ represents the perceptual distance of the agent $i$.

**Case (b):** In this case, a subset of agents forms clusters, while agent $i$ has no neighbors within its perceptual range. To establish the aforementioned tacit consensus, agent $i$ should move toward its nearest agent. Accordingly, the *advantageous configuration* $C_{\mathrm{adv}}^t$ is defined as the minimum distance between agent $i$ and its nearest agent, and the *agent-specific configuration* $C_i^t$ denotes the distance between agent $i$ and its nearest agent. Formally, $C_{\mathrm{adv}}^t$ and $C_i^t$ are defined as:

$$\begin{cases} C_i^t = \left\| \varphi_i^t - \varphi_k^t \right\|_2 \\ C_i^{t+1} = \left\| \varphi_i^{t+1} - \varphi_k^t \right\|_2 \\ C_{\mathrm{adv}}^t = \left\| \varphi_{k\_\mathrm{min}}^t - \varphi_k^t \right\|_2, \end{cases} \tag{16}$$

where $\varphi_k^t$ denotes the position of agent $i$'s nearest agent $k$, $\varphi_{k\_\min}$ refers to the position with the minimum distance to agent $k$'s position.

The parameter term $\lambda^{i,t}$ of case (b) is defined in 14. The tacit reward term $r_{\mathrm{tac}}^{i,t}$ is defined in 17.

$$r_{\mathrm{tac}}^{i,t} = \lambda_b^{i,t} \times \left( \left\| C_i^t - C_{\mathrm{adv}}^t \right\| - \left\| C_i^{t+1} - C_{\mathrm{adv}}^t \right\| \right). \tag{17}$$

**Case (c):** In this case, agent $i$ perceives more than one other agent within its perceptual range. To establish the aforementioned tacit consensus, agent $i$ is expected to continuously detect neighboring agents while maintaining a specified inter-agent distance. Accordingly, the ***advantageous configuration*** $C_{\mathrm{adv}}^t$ is defined as the target distance between agent $i$ and the position that maintains the specified inter-agent distance relative to its nearest neighbor, and the ***agent-specific configuration*** $C_i^t$ denotes the distance between agent $i$ and its nearest agent. Formally, $C_{\mathrm{adv}}^t$ and $C_i^t$ are defined as:

$$\begin{cases} C_i^t = \left\| \varphi_i^t - \varphi_k^t \right\|_2 \\ C_i^{t+1} = \left\| \varphi_i^{t+1} - \varphi_k^t \right\|_2 \\ C_{\mathrm{adv}}^t = \left\| \varphi_{k\_\mathrm{sd}}^t - \varphi_k^t \right\|_2, \end{cases} \tag{18}$$

where $\varphi_{k\_\mathrm{sd}}^t$ denotes the position at which agent $i$ maintains the specified distance relative to its nearest agent $k$.

The parameter term $\lambda^{i,t}$ of case (c) is defined in 19. The tacit reward term $r_{\mathrm{tac}}^{i,t}$ is defined in 20.

$$\lambda_c^{i,t} = \begin{cases} \dfrac{\min\limits_{k \in \rho_i} \left\| \varphi_i - \varphi_k \right\|_2}{\alpha d_k}, & \text{if } \min\limits_{k \in \rho_i} \left\| \varphi_i - \varphi_k \right\|_2 < \alpha d_k \\[4mm] \dfrac{\max\limits_{k \in \rho_i} \left\| \varphi_i - \varphi_k \right\|_2 - \alpha d_k}{d_k - \alpha d_k}, & \text{if } \alpha d_k \le \min\limits_{k \in \rho_i} \left\| \varphi_i - \varphi_k \right\|_2 < d_k, \end{cases} \tag{19}$$

where $\alpha$ is a parameter in the range $(0, 1)$, representing a specified distance relative to agent $k$'s perceptual range $d_k$.

$$r_{\mathrm{tac}}^{i,t} = \lambda_c^{i,t} \times \left( \left\| C_i^t - C_{\mathrm{adv}}^t \right\| - \left\| C_i^{t+1} - C_{\mathrm{adv}}^t \right\| \right). \tag{20}$$

**Case (d):** In this case, the agent perceive only one other agent within its perceptual range. Agents in this case are classified as either leader agents or follower agents, based on their relative positions to the nearby agents. We designate the agent positioned on the left as the leader, while the other is assigned as the follower. The corresponding subscripts for the leader and follower agents are denoted as $l$ and $f$ respectively.

To establish the aforementioned tacit consensus, the leader agent is expected to move toward the nearest agent outside its perceptual range. Accordingly, the ***advantageous configuration*** of the leader agent, denoted as $C_{\mathrm{adv\_l}}^t$, is defined as the minimum distance between agent $l$ and its nearest agent outside its perceptual range. The ***agent-specific configuration*** of the leader agent $C_l^t$ represents the distance between agent $l$ and its nearest agent outside its perceptual range. Formally, $C_{\mathrm{adv\_l}}^t$ and $C_l^t$ are defined as:

$$\begin{cases} C_l^t = \left\| \varphi_l^t - \varphi_k^t \right\|_2 \\ C_l^{t+1} = \left\| \varphi_l^{t+1} - \varphi_k^t \right\|_2 \\ C_{\mathrm{adv\_l}}^t = \left\| \varphi_{k\_\min}^t - \varphi_k^t \right\|_2, \end{cases} \tag{21}$$

where $\varphi_k^t$ denotes the position of agent $k$, which is the nearest agent outside the perceptual range of agent $l$, and $\varphi_{k\_\min}$ denotes the position that is closest to agent $k$.

The parameter term $\lambda_d^{l,t}$ for leader agent in case (d) is defined in 22. The tacit reward term $r_{\mathrm{tac}}^{l,t}$ is defined in 23.

$$\lambda_d^{l,t} = \min\left\{\frac{\min\limits_{k\notin\rho_l^t}\|\varphi_l^t - \varphi_k^t\|_2 - d_l}{\beta d_l - d_l}, 1\right\}, \tag{22}$$

$$r_{\text{tac}}^{l,t} = \lambda_d^{l,t} \times \left(\|C_l^t - C_{\text{adv\_}l}^t\| - \|C_l^{t+1} - C_{\text{adv\_}l}^t\|\right). \tag{23}$$

To establish the aforementioned tacit consensus, the follower agent is expected to move toward the leader agent. Accordingly, the ***advantageous configuration*** of the follower agent, denoted as $C_{\text{adv\_}f}^t$, is defined as the minimum distance between agent $f$ and agent $l$. The ***agent-specific configuration*** of the follower agent $C_f^t$ represents the distance between agent $f$ and agent $l$. Formally, $C_{\text{adv\_}f}^t$ and $C_f^t$ are defined as:

$$\begin{cases} C_f^t = \left\|\varphi_f^t - \varphi_l^t\right\|_2 \\ C_f^{t+1} = \left\|\varphi_f^{t+1} - \varphi_l^t\right\|_2 \\ C_{\text{adv\_}f}^t = \|\varphi_{l\_\min}^t - \varphi_l^t\|_2, \end{cases} \tag{24}$$

where $\varphi_{l\_\min}$ denotes the position that is closest to agent $l$.

The parameter term $\lambda_d^{f,t}$ for follower agent in case (d) is defined in 25. The tacit reward term $r_{\text{tac}}^{f,t}$ is defined in 26.

$$\lambda_d^{f,t} = \begin{cases} \dfrac{\left\|\varphi_l^t - \varphi_f^t\right\|_2}{\alpha d_f}, \text{ if } \left\|\varphi_l^t - \varphi_f^t\right\|_2 < \alpha d_f \\[3mm] \dfrac{\left\|\varphi_l^t - \varphi_f^t\right\|_2 - \alpha d_f}{d_f - \alpha d_f}, \text{ if } \alpha d_f \leq \left\|\varphi_l^t - \varphi_f^t\right\|_2 < d_f, \end{cases} \tag{25}$$

$$r_{\text{tac}}^{f,t} = \lambda_d^{f,t} \times \left(\left\|C_f^t - C_{\text{adv\_}f}^t\right\| - \left\|C_f^{t+1} - C_{\text{adv\_}f}^t\right\|\right). \tag{26}$$

## B.2 PREDATOR-PREY

For Predator-Prey task, our intuitive insight is that agents can achieve the overall objective more efficiently if they move closer to each other. Considering the partial observability in Predator-Prey environments, we define the ***tacit consensus*** among agents as agents intentionally reducing their pairwise distances. To facilitate the efficient acquisition of the desired tacit behavior, we categorize three representative cases and define the corresponding ***advantageous configuration*** $C_{\text{adv}}$ and ***agent-specific configuration*** $C_i^t$, where agent $i$ represents a typical agent among multiple agents. Additionally, a parameter term $\lambda^{i,t}$ is introduced prior to the tacit reward to dynamically adjust its weight during training, thereby improving learning efficiency.

**Case (a):** In this case, agent $i$ has no neighbors within its perceptual range. To establish the aforementioned tacit consensus, agent $i$ should move toward its nearest agent. Accordingly, the ***advantageous configuration*** $C_{\text{adv}}^t$ is defined as the minimum distance between agent $i$ and its nearest agent, and the ***agent-specific configuration*** $C_i^t$ denotes the distance between agent $i$ and its nearest agent. Formally, $C_{\text{adv}}^t$ and $C_i^t$ are defined as:

$$\begin{cases} C_i^t = \|\varphi_i^t - \varphi_k^t\|_2 \\ C_i^{t+1} = \|\varphi_i^{t+1} - \varphi_k^t\|_2 \\ C_{\text{adv}}^t = \|\varphi_{k\_\min}^t - \varphi_k^t\|_2, \end{cases} \tag{27}$$

where $\varphi_k^t$ denotes the position of agent $i$'s nearest agent $k$, $\varphi_{k\_\min}$ refers to the position with the minimum distance to agent $k$'s position.

The parameter term $\lambda_a^{i,t}$ of case (a) is defined in 28. The tacit reward term $r_{\text{tac}}^{i,t}$ is defined in 29.

$$\lambda_a^{i,t} = \min\left\{ \frac{\|\varphi_i^t - \varphi_k^t\|_2 - (d_i + 1)}{\sqrt{(d_i + 1)^2 + (d_i + 2)^2} - (d_i + 1)}, 1 \right\}, \tag{28}$$

$$r_{\text{tac}}^{i,t} = \lambda_a^{i,t} \times \left( \left\|C_i^t - C_{\text{adv}}^t\right\| - \left\|C_i^{t+1} - C_{\text{adv}}^t\right\| \right), \tag{29}$$

where the term $d_i$ represents the perceptual distance of the agent $i$.

**Case (b):** In this case, agent $i$ perceives more than one other agent within its perceptual range. To establish the aforementioned tacit consensus, agent $i$ is expected to approach its nearest neighbor as closely as possible. Accordingly, the ***advantageous configuration*** $C_{\text{adv}}^t$ is defined as the minimum achievable distance to agent $i$'s nearest agent, and the ***agent-specific configuration*** $C_i^t$ denotes the distance between agent $i$ and its nearest agent. The formal definitions of $C_{\text{adv}}^t$ and $C_i^t$ are defined as 27.

The parameter term $\lambda_b^{i,t}$ of case (b) is defined in 30. The tacit reward term $r_{\text{tac}}^{i,t}$ is defined in 31.

$$\lambda_b^{i,t} = \begin{cases} 0, & \text{if } \|\varphi_i^t - \varphi_k^t\|_2 = 1 \\ \dfrac{d_i - \|\varphi_i^t - \varphi_k^t\|_2}{d_i - 1}, & \text{if } 1 < \|\varphi_i^t - \varphi_k^t\|_2 \le d_i \\ 1, & \text{if } d_i < \|\varphi_i^t - \varphi_k^t\|_2, \end{cases} \tag{30}$$

$$r_{\text{tac}}^{i,t} = \lambda_b^{i,t} \times \left( \left\|C_i^t - C_{\text{adv}}^t\right\| - \left\|C_i^{t+1} - C_{\text{adv}}^t\right\| \right), \tag{31}$$

where $\varphi_k^t$ denotes the position of agent $i$'s nearest agent $k$.

**Case (c):** In this case, the agent perceives only one other agent within its perceptual range. Agents in this case are classified as either leader agents or follower agents, based on their relative positions to the nearby agents. We designate the agent positioned on the left as the leader, while the other is assigned as the follower. The corresponding subscripts for the leader and follower agents are denoted as $l$ and $f$ respectively.

To establish the aforementioned tacit consensus, the leader agent is expected to move toward the nearest agent outside its perceptual range. Accordingly, the ***advantageous configuration*** of the leader agent, denoted as $C_{\text{adv\_}l}^t$, is defined as the minimum distance between agent $l$ and its nearest agent outside its perceptual range. The ***agent-specific configuration*** of the leader agent $C_l^t$ represents the distance between agent $l$ and its nearest agent outside its perceptual range. Formally, $C_{\text{adv\_}l}^t$ and $C_l^t$ are defined as 21.

The parameter term $\lambda_c^{l,t}$ of case (c) is defined in 32. The tacit reward term $r_{\text{tac}}^{l,t}$ is defined in 33.

$$\lambda_c^{l,t} = \min\left\{ \frac{\min\limits_{k \notin \rho_l^t} \|\varphi_l^t - \varphi_k^t\|_2 - (d_l + 1)}{\sqrt{(d_l + 1)^2 + (d_l + 2)^2} - (d_l + 1)}, 1 \right\}, \tag{32}$$

$$r_{\text{tac}}^{l,t} = \lambda_c^{l,t} \times \left( \left\|C_l^t - C_{\text{adv\_}l}^t\right\| - \left\|C_l^{t+1} - C_{\text{adv\_}l}^t\right\| \right), \tag{33}$$

where the set $\rho_l$ denotes the collection of agents within the perceptual range of the agent $l$.

To establish the aforementioned tacit consensus, the follower agent is expected to move toward the leader agent. Accordingly, the ***advantageous configuration*** of the follower agent, denoted as $C_{\text{adv\_}f}^t$, is defined as the minimum distance between agent $f$ and agent $l$. The ***agent-specific configuration*** of the follower agent $C_f^t$ represents the distance between agent $f$ and agent $l$. Formally, $C_{\text{adv\_}f}^t$ and $C_f^t$ are defined as 24.

The parameter term $\lambda_c^{f,t}$ for follower agent in case (c) is defined in 34. The tacit reward term $r_{\text{tac}}^{f,t}$ is defined in 35.

$$\lambda_c^{f,t} = \begin{cases} 0, & \text{if } \left\| \varphi_f^t - \varphi_l^t \right\|_2 = 1 \\ \dfrac{d_f - \left\| \varphi_f^t - \varphi_l^t \right\|_2}{d_f - 1}, & \text{if } 1 < \left\| \varphi_f^t - \varphi_l^t \right\|_2 \le d_f \\ 1, & \text{if } d_f < \left\| \varphi_f^t - \varphi_l^t \right\|_2, \end{cases} \tag{34}$$

$$r_{\text{tac}}^{f,t} = \lambda_c^{f,t} \times \left( \left\| C_f^t - C_{\text{adv\_}f}^t \right\| - \left\| C_f^{t+1} - C_{\text{adv\_}f}^t \right\| \right). \tag{35}$$

## C ENVIRONMENTS AND IMPLEMENTATION DETAILS

### C.1 DETAILED EXPERIMENTAL SETUP

All experiments in this paper are run on Nvidia GeForce RTX 3090 graphics cards and AMD EPYC 7H12 64-Core processor CPU. To ensure fair comparisons, we fine-tuned the hyperparameters of all baseline models accordingly. In SMAC environments, PTMC and all baseline methods in our paper are trained in the same testbeds. In addition, some methods involve specific parameter tuning across different scenarios, making comparisons unfair. Therefore, based on relevant literature, we adjusted the parameters of these methods appropriately for each scenario and ensured consistent parameters across different random seeds within same scenario during our experiments.

### C.2 SMAC

Table 1: SMAC maps in different scenarios.

| Name | Ally Units | Enemy Units | Type | Difficulty |
|---|---|---|---|---|
| 3s_vs_5z | 3 Stalkers | 5 Zealots | micro-trick: kiting | Hard |
| corridor | 6 Zealots | 24 Zerglings | micro-trick: wall off | Super Hard |
| 6h_vs_8z | 6 Hydralisks | 8 Zealots | micro-trick: focus fire | Super Hard |
| 2s3z | 2 Stalkers & 3 Zealots | 2 Stalkers & 3 Zealots | heterogeneous & symmetric | Easy |
| 2s_vs_1sc | 2 Stalkers | 1 Spine Crawler | micro-trick: alternating fire | Hard |
| 5m_vs_6m | 5 Marines | 6 Marines | homogeneous & asymmetric | Hard |
| 3s5z_vs_3s6z | 3 Stalkers & 5 Zealots | 3 Stalkers & 6 Zealots | heterogeneous & asymmetric | Super Hard |

In SMAC tasks, a group of units controlled by decentralized agents cooperates to defeat the enemy agent system controlled by handcrafted heuristics. Each agent has its perceptual range, and at each timestep the agent can observe information about allied and enemy units within that range, including distance, relative position, health, shield amount, and unit type. The global state information, which includes all agents' positions, health, allied units' previous actions, and cooldowns, is only accessible during centralized training. The objective of our agents is to defeat enemy agents within a limited timesteps, with the environment's reward function tied to the health of both units. Battles can be either symmetric or asymmetric, and the group of agents can be homogeneous or heterogeneous. Asymmetric scenarios generally present a higher level of learning difficulty. Additionally, scenarios with larger numbers of agents tend to pose greater challenges for coordinated learning. Table 1 provides a detailed description of each SMAC scenario.

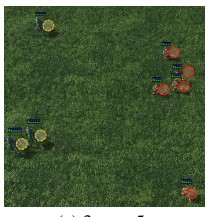 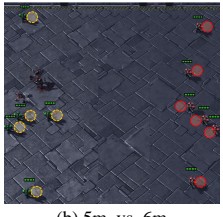 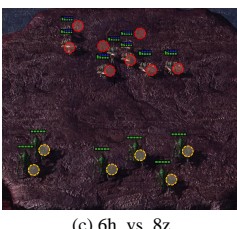

(a) 3s_vs_5z        (b) 5m_vs_6m        (c) 6h_vs_8z

Figure 8: The thumbnails of initial position in 3s_vs_5z, 5m_vs_6m and 6h_vs_8z.

In this paper, **we increase the difficulty of the original SMAC maps by randomizing the initial positions of both sides and creating scenarios in which agents cannot initially perceive one another**. For clarity, Figure 8 presents thumbnails of the modified maps, where red circles denote enemy units and yellow circles denote allied units. **The modified map files (.SC2Map) are provided in the supplementary material.**

## C.3 SMACv2

SMACv2 is an updated version of SMAC, introducing increased variability and difficulty through randomized start positions and unit types. Specifically, start positions are randomized in two ways: (a) *Reflect scenario*, where allied positions are randomly assigned, and enemy positions are symmetrically reflected across the map's midpoint (as shown in Figure 9(a)); and (b) *Surround scenario*, where allied units spawn in the center of the map and are encircled by enemy units (as shown in Figure 9(b)). The probability of each scenario type is controlled by a parameter $p \in [0,1]$.

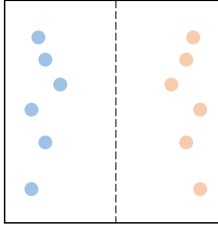 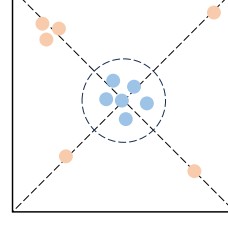

(a) The reflect scenario        (b) The surround scenario

Figure 9: Illustration of two starting position types: reflect and surround. Allied units are shown in blue; enemy units in orange.

Moreover, unlike SMAC where unit types are fixed, SMACv2 introduces randomized unit compositions based on predefined generation probabilities (Table 2). For each StarCraft II race (Protoss, Terran, and Zerg), three unit types are selected. For example, in the `Zerg_10_vs_10` scenario, both the 10 allied and 10 enemy units are independently sampled from the Zerg race according to the predefined generation probabilities.

Table 2: Unit types and generation probabilities for each race in SMACv2.

| Race | Unit Types | Generation Probabilities |
|---|---|---|
| Terran | Marine, Marauder, Medivac | 0.45, 0.45, 0.10 |
| Protoss | Stalker, Zealot, Colossus | 0.45, 0.45, 0.10 |
| Zerg | Zergling, Hydralisk, Baneling | 0.45, 0.45, 0.10 |

## C.4 PREDATOR-PREY

In the Predator-Prey task, predator agents attempt to capture two types of prey—stags and hares—within a grid-based environment. Each predator has a local observation radius of three grids centered on itself, within which it can perceive the relative positions of other agents and nearby prey,

as well as the prey type. Prey do not move proactively; instead, they respond only to blocked movements after the predators have acted. The global state includes the relative positions of all agents and prey across the entire grid. Both predators and prey can move to one of the four adjacent grid or remain stationary. Movements are executed sequentially: predators move first in random order, followed by the prey selecting a random valid action (i.e. an action that would not lead to a collision with another entity). A prey is considered captured if a sufficient number of predator agents occupy the adjacent grids and perform a capture action. Specifically, capturing a stag requires at least two predators to simultaneously occupy adjacent grids, while capturing a hare requires at least one. Due to differing capture difficulty, rewards vary: successfully capturing a stag yields a shared reward of +10, whereas capturing a hare yields +2. Consequently, predator agents are expected to prioritize coordinated efforts to capture stags rather than acting independently. To intuitively illustrate the Predator-Prey tasks, we visualize a simplified example. Specifically, we consider a scenario where five agents (predators) aim to capture four stags and three hares within a 6×6 grid , as shown in Figure 10.

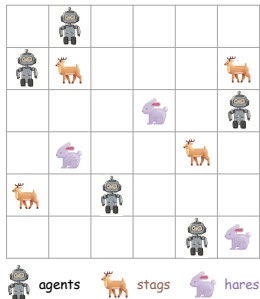

Figure 10: Illustration of a Predator-Prey task in a 6×6 grid (5 agents, 4 stags, 3 hares).

# D    RESULTS FOR PRE-TRAINING

In the main text, we mention that during the tacit pre-training phase, the tacit metric $M_{\text{tac}}$ is monitored to evaluate the emergence of tacit behavior. When it exceeds a predefined threshold $M_{\text{tac}}^*$, the pre-training process is terminated, indicating that the agents have acquired the coordination capability necessary for forming tacit behavior. The threshold $M_{\text{tac}}^*$ is adjusted according to different experimental scenarios. In addition, based on our previous definition of tacit reward, we design distinct tacit rewards for different cases. Consequently, each case is associated with a specific tacit metric and its corresponding threshold $M_{\text{tac}}^*$. During training, we consider the extent to which the tacit metric in each case reaches its designated threshold, and terminate the pre-training accordingly. For instance, in the SMAC environment under the `3s_vs_5z` scenario, we empirically set the thresholds for the four cases to [0.75, 0.85, 0.85, 0.85], and the pre-training process is terminated once all four conditions are satisfied. Specifically, for the `6h_vs_8z` and `corridor` scenarios, the thresholds for the four cases are set to [0.9, 0.85, 0.9, 0.9] and [0.75, 0.8, 0.85, 0.8], respectively. Furthermore, we report the results of ten independent pre-training runs, presenting the corresponding tacit metric values of the four cases at the point when pre-training is terminated for each scenario. The statistical results are shown in Figure 11, where the height of each bar represents the mean value, with the exact values labeled on the bars. The error bars represent the corresponding standard deviation.

For the predator-prey environment, we empirically set the thresholds for the three cases to [0.75, 0.9, 0.85]. Similarly, we report the results of ten independent pre-training runs, presenting the corresponding tacit metric values of the three cases at the point when pre-training is terminated. The statistical results are illustrated in Figure 12.

As mentioned in the main text, unlike conventional pre-training, our tacit pre-training focuses on simpler individual-agent tasks and requires significantly fewer training steps compared to coordinated training. To further illustrate the differenc in training steps between the two training phases, we report the average number of training steps at the end of tacit pre-training across 10 independent

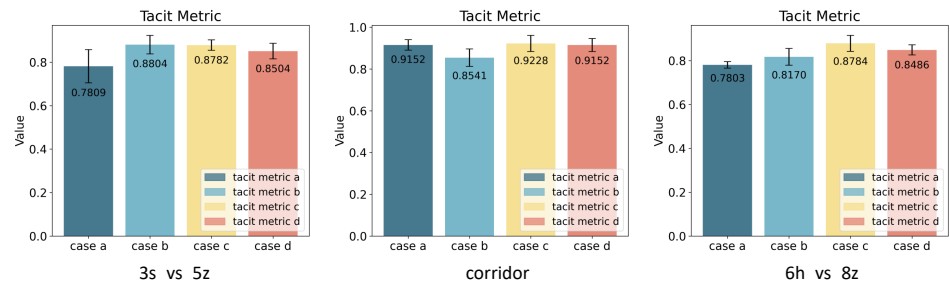

Figure 11: Mean and standard deviation of tacit metrics across four cases in three SMAC tasks.

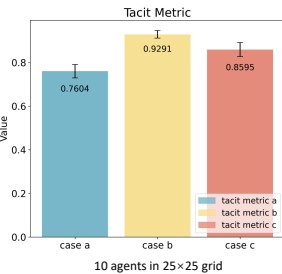

Figure 12: Mean and standard deviation of tacit metrics across three cases in Predator-Prey task.

runs for three scenarios in the SMAC environment, along with the corresponding standard deviations (std). The detailed results are summarized in Table 3.

Table 3: Training steps (mean ± std.) of tacit pre-training in three SMAC scenarios.

| Scenario | 3s_vs_5z | corridor | 6h_vs_8z |
|---|---|---|---|
| Training Steps ($\times 10^3$) | 588.8 ± 114.5 | 871.7 ± 120.3 | 394.1 ± 111.4 |

For the SMAC tasks, coordinated training involves 6 million training steps, which is approximately ten times the number required for tacit pre-training. In the predator-prey environment, we evaluate the average number of training steps at the end of tacit pre-training over 10 independent runs, yielding a mean of 33.2k steps with a standard deviation of 12.5k. In contrast, coordinated training in this environment requires 500k steps, which is over ten times the number of steps required for tacit pre-training. These results clearly highlight the distinction between the PTMC framework and the conventional paradigm of extensive pre-training followed by light fine-tuning, demonstrating the greater learning efficiency of PTMC in MARL.

# E  ADDITIONAL RESULTS FOR COMPARATIVE EVALUATION

To strengthen the "Comparative Evaluation" section in main text, we further incorporate diverse scenarios from Predator-Prey, SMAC, and SMACv2 environments to evaluate the effectiveness of PTMC across a broader range of tasks.

## E.1  COMPARATIVE PERFORMANCE EVALUATION ON PREDATOR-PREY TASKS

In the main text, Table 1 reports the mean and standard deviation of returns after 500k steps across three Predator-Prey scenarios. In this section, we present detailed training curves and further experiments under various settings, including different opponent combinations and grid configurations. For clarity, we compare MAPPO-based and QMIX-based algorithms separately. In the following figures, solid lines represent the mean over five random seeds, and shaded areas indicate the corresponding confidence intervals.

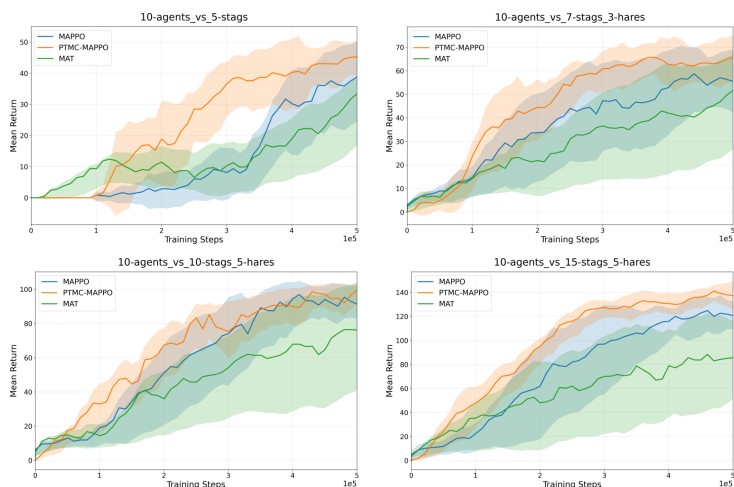

Figure 13: Comparative training performance of MAPPO-based algorithms in predator-prey (25×25 grid).

As shown in Figure 13, all experiments employ the same pre-trained tacit model to facilitate coordination among ten agents in a 25×25 grid. Across scenarios with varying prey numbers (more or fewer than predators) and different compositions of stags and hares, PTMC-MAPPO consistently outperforms both MAPPO and MAT in terms of cooperative performance. MAPPO generally ranks second, though it demonstrates limited training efficiency in the early stages. MAT yields the lowest final mean return across all four scenarios. Despite initially outperforming both PTMC and MAPPO in the "10 agents vs 5 stags" scenario, it fails to sustain this early advantage. Additionally, MAT exhibits considerable training instability, as reflected by large confidence intervals. These results highlight the scalability of the tacit pre-training mechanism and further substantiate the effectiveness of PTMC.

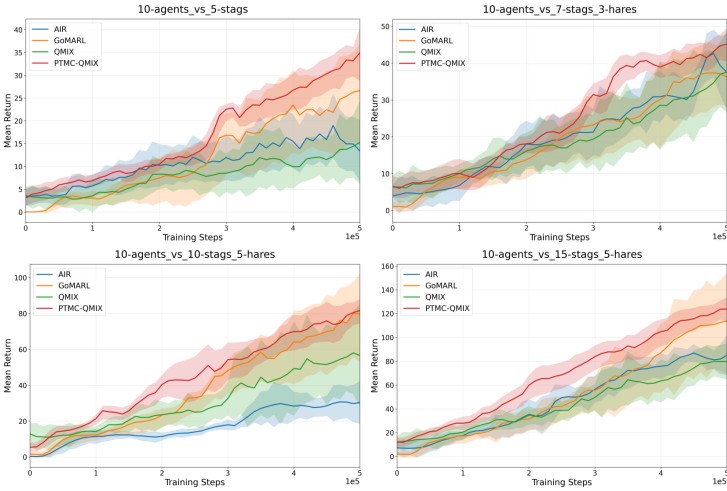

Figure 14: Comparative training performance of QMIX-based algorithms in predator-prey (25×25 grid).

We further evaluate PTMC-QMIX in the same scenarios, using QMIX as the base algorithm, and compare its performance with three QMIX-based variants, as shown in Figure 14. All methods utilize the same pre-trained tacit model. Across all four scenarios, PTMC-QMIX consistently achieves superior final mean returns. In the "10 agents vs 5 stags" and "10 agents vs 15 stags and 5 hares" settings, AIR exhibits faster early-stage learning but is slightly outperformed by PTMC-QMIX in

terms of final return. In the "10 agents vs 10 stags and 5 hares" scenario, AIR demonstrates clearly inferior overall training efficiency, further underscoring the robustness and performance advantage of PTMC-QMIX. GoMARL achieves comparable final returns to PTMC-QMIX in the "10 agents vs 10 stags and 5 hares" and "10 agents vs 15 stags and 5 hares" scenarios, although its early-stage learning is slightly slower. As a classical baseline, QMIX exhibits stable training but consistently shows lower efficiency and suboptimal final performance across all scenarios. These results further demonstrate the flexibility of PTMC, which can be incorporated into both QMIX- and MAPPO-based frameworks while consistently improving performance.

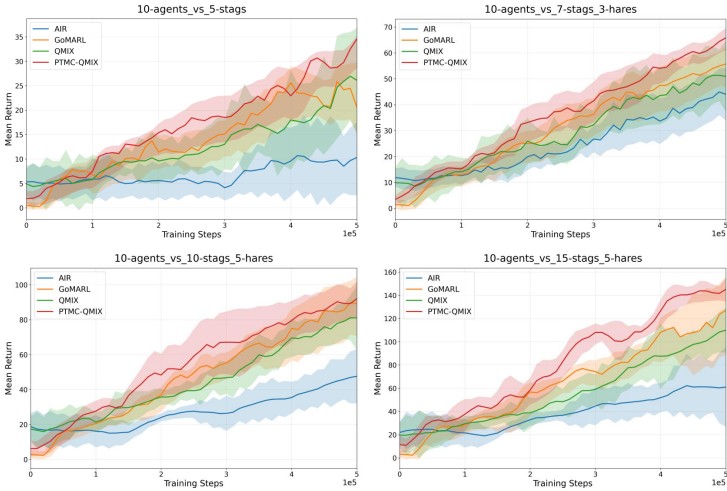

Figure 15: Comparative training performance of QMIX-based algorithms in predator-prey (20×20 grid).

Moreover, we evaluate the training performance of PTMC under different grid sizes in predator-prey tasks. Specifically, PTMC-QMIX and three QMIX-based baselines are tested across four scenarios on a 20×20 grid, as shown in Figure 15. Overall, PTMC-QMIX consistently achieves the best performance across all scenarios, both in terms of learning speed and final mean return. Among the baselines, GoMARL and QMIX exhibit similar trends, with GoMARL showing slightly better overall performance. Notably, while AIR demonstrate competitive performance with PTMC-QMIX in some 25×25 grid settings (Figure 14), it performs significantly worse across all four scenarios in the 20×20 grid settings, both in learning efficiency and final performance. These results further confirm that PTMC maintains stable performance across diverse predator-prey scenarios, and that the tacit pre-training mechanism exhibits robust and effective scalability.

### E.2 COMPARATIVE PERFORMANCE EVALUATION ON SMAC TASKS

To evaluate the generality of our approach, we further include a range of SMAC scenarios with varying difficulty levels and type diversity, including homogeneous or heterogeneous agents, symmetric or asymmetric setups, and distinct micro-tracks. We conduct comparisons between PTMC and their respective QMIX-based and MAPPO-based baselines.

As shown in Figure 16, we compare PTMC-QMIX with three QMIX-based baselines across three SMAC scenarios. PTMC-QMIX achieves the highest final win rates on all maps. On `2s_vs_1sc`, QMIX exhibits faster early-stage learning, and in `5m_vs_6m`, all baselines show more rapid initial improvement. However, PTMC-QMIX consistently outperforms them in terms of final performance on both maps. On the super hard map `3s5z_vs_3s6z`, PTMC-QMIX demonstrates clear advantages in both learning efficiency and final win rate. Overall, PTMC-QMIX exhibits strong performance across diverse SMAC tasks.

To further validate the effectiveness of PTMC on SMAC, we compare PTMC-MAPPO with MAPPO and MAT across three scenarios, as shown in Figure 17. Although MAT exhibits a slight advantage over PTMC-MAPPO in `5m_vs_6m`, PTMC-MAPPO still significantly outperforms MAPPO, upon which it is based. In the other two scenarios, PTMC-MAPPO consistently demonstrates superior

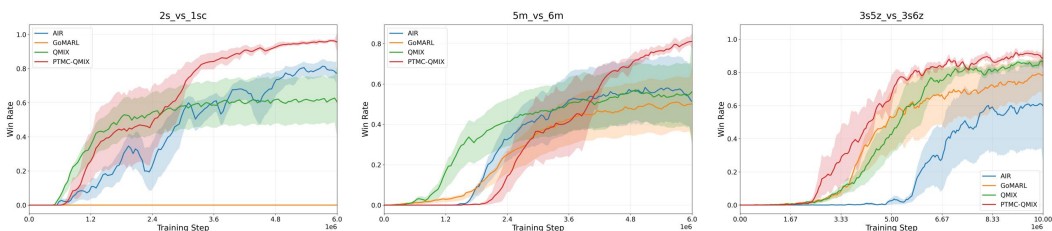

Figure 16: Training performance comparison of QMIX-based algorithms in SMAC.

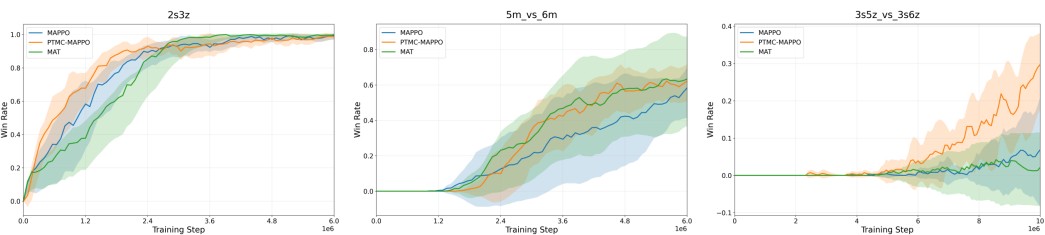

Figure 17: Training performance comparison of MAPPO-based algorithms in SMAC.

performance. Specifically, on `2s3z`, it achieves a steeper improvement curve; and on the super hard map `3s5z_vs_3s6z`, it shows a substantial advantage, whereas MAPPO and MAT exhibit only marginal gains in test win rates. These results further confirm the consistent superiority of PTMC when built upon either QMIX or MAPPO, highlighting its effectiveness across diverse scenarios.

### E.3    COMPARATIVE PERFORMANCE EVALUATION ON SMACV2 TASKS

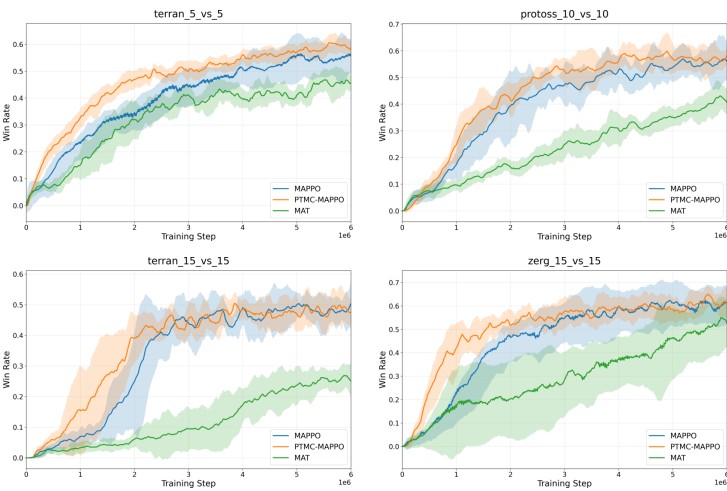

Figure 18: Training performance comparison of MAPPO-based algorithms in SMACv2.

To complement the three scenarios presented in Figure 4 of the main text, we further evaluate PTMC-MAPPO against MAPPO and MAT in four SMACv2 scenarios, as shown in Figure 18. PTMC-MAPPO consistently achieves superior performance across all four scenarios. Although MAPPO exhibits comparable final performance in `terran_15_vs_15`, it demonstrates a slower learning curve compared to PTMC-MAPPO. In contrast, MAT consistently underperforms, yielding the low-

est win rates in all scenarios. These additional results on SMACv2 further validate the robustness and effectiveness of PTMC across a wider range of tasks.

# F  ADDITIONAL RESULTS FOR ABLATION STUDY

We conduct ablation studies on three SMAC scenarios to evaluate the contributions of key components in PTMC, where PTMC is built on MAPPO. Specifically, "PTMC w/o Constr." removes the tacit constraint term; "PTMC w/o Pretr." omits actor network initialization from tacit pre-training, while still allowing selective alignment with the pre-trained tacit model during coordinated training; and "PTMC w/o BinGate." removes the binary gating function to assess the effect of selective constraint enforcement.

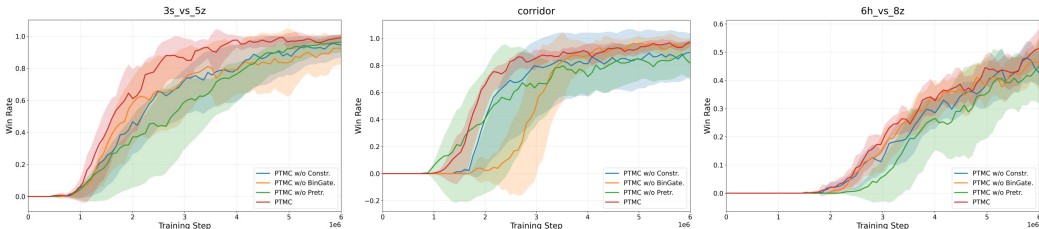

Figure 19: Ablation study of PTMC in SMAC scenarios.

As shown in Figure 19, PTMC consistently outperforms all ablated variants. Notably, "PTMC w/o Pretr." performs significantly worse across all three scenarios, particularly exhibiting a slower initial performance increase and poorer final coordinated performance in 6h_vs_8z and corridor, underscoring the importance of pre-trained initialization. In addition, PTMC demonstrates a clear early-stage advantage over "PTMC w/o Constr." in all three scenarios, validating the benefit of incorporating the tacit constraint term during coordinated training. Furthermore, while "PTMC w/o BinGate." achieves performance comparable to PTMC in 6h_vs_8z and remains the second-best performer in 3s_vs_5z, it suffers a noticeable early-stage performance drop in the corridor scenario. This may be attributed to the adverse effect of indiscriminate constraint enforcement, which could impair learning due to inaccurate loss estimation.

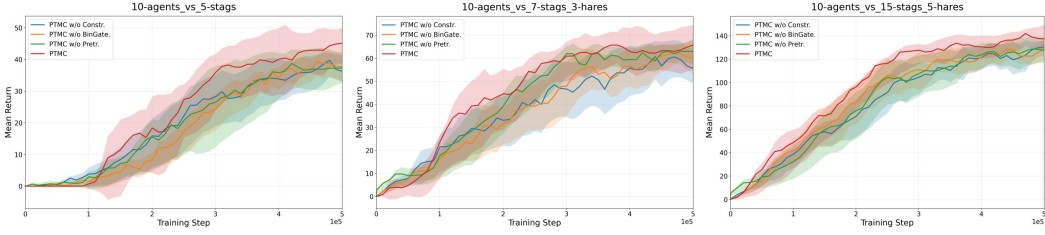

Figure 20: Ablation study of PTMC in Predator-Prey scenarios.

We also conduct ablation studies in the Predator-Prey environment, as shown in Figure 20. PTMC consistently outperforms all ablated variants across the three predator-prey scenarios. "PTMC w/o Constr." performs poorly in all cases, highlighting the effectiveness of aligning with the pre-trained tacit model during coordinated training. Compared to its performance in SMAC, the greater impact of the tacit constraint term in this environment can be attributed to the higher relevance between the designed tacit reward and the task objective in Predator-Prey. A similar performance degradation is observed for "PTMC w/o BinGate.", where inaccurate loss estimation likely hinders effective learning, resulting in inferior performance compared to full PTMC. Interestingly, unlike the SMAC results, "PTMC w/o Pretr." achieves the suboptimal performance in the "10 agents vs 7 stags and

3 hares" scenario. This suggests that the effect of pre-trained initialization is relatively limited in this task, while the tacit constraint term still plays a critical role in guiding coordination during coordinated training.

## G EXPERIMENTAL EVALUATION OF KEY PARAMETER SETTINGS

In coordinated training, a tacit constraint term $\mathcal{L}_{\text{tac}}$ is incorporated into the loss function, as defined in Eq 36. The hyperparameter $\alpha_{\text{tac}}$ is introduced as a weighting coefficient to balance the contribution of $\mathcal{L}_{\text{tac}}$ relative to the main loss term $\mathcal{L}_{\text{main}}$.

$$J(\theta_{\text{coor}}) = \mathcal{L}_{\text{main}} - \alpha_{\text{tac}} \cdot \mathcal{L}_{\text{tac}}. \tag{36}$$

To ensure that the magnitude of the tacit constraint term remains within a stable range relative to the main loss term throughout training, we design the hyperparameter $\alpha_{\text{tac}}$ as an adaptive coefficient, defined as:

$$\alpha_{\text{tac}} = \alpha \cdot \alpha_{\text{adapt}}, \tag{37}$$

where $\alpha$ denotes a base coefficient representing the desired order-of-magnitude difference between $\mathcal{L}_{\text{tac}}$ and $\mathcal{L}_{\text{main}}$, and $\alpha_{\text{adapt}}$ is an adaptive scaling factor that dynamically adjusts $\mathcal{L}_{\text{tac}}$ to match the scale of $\mathcal{L}_{\text{main}}$ during training. Specifically, $\alpha_{\text{adapt}}$ is computed by first calculating the ratio between the main loss $\mathcal{L}_{\text{main}}$ and the value of the tacit constraint term $\mathcal{L}_{\text{tac}}$:

$$\text{ratio} = \frac{\mathcal{L}_{\text{main}}}{\mathcal{L}_{\text{tac}}}. \tag{38}$$

Then, $\alpha_{\text{adapt}}$ is selected as the closest value to this ratio from a predefined set of scaling candidates:

$$\alpha_{\text{adapt}} = \arg\min_{x \in \mathcal{X}} |\text{ratio} - x|, \tag{39}$$

where $\mathcal{X} = 10^{-i} \mid i \in \mathbb{Z}$ denotes the set of candidate scaling factors in descending order of magnitude.

To validate the effectiveness and robustness of the proposed adaptive weighting scheme, we conduct a series of experiments in both the SMAC and Predator-Prey environments, where PTMC is built on MAPPO. Specifically, we evaluate the impact of different base coefficient values $\alpha$ across three representative scenarios in each environment. These experiments aim to investigate whether the overall training dynamics and coordinated performance are sensitive to the choice of $\alpha$, and to determine an appropriate magnitude range for this hyperparameter. The analysis helps assess the stability of our adaptive mechanism and its ability to maintain effective regularization throughout training without requiring fine-tuned manual adjustments.

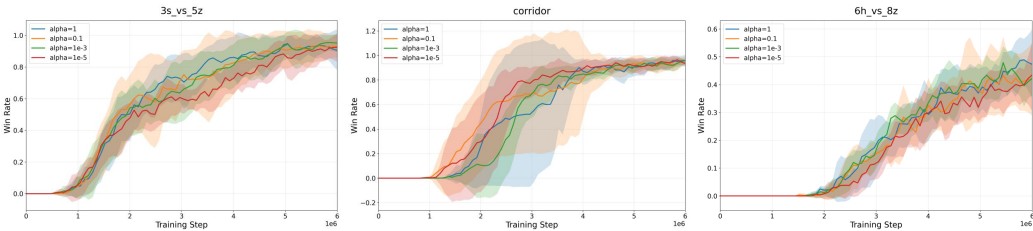

Figure 21: Effect of $\alpha$ parameter on optimization in SMAC tasks.

As illustrated in Figure 21, the choice of the $\alpha$ parameter has a limited impact on the final coordinated performance across most scenarios. For instance, in the `3s_vs_5z` and `corridor` tasks—both of which exhibit convergence—different values of $\alpha$ yield similar final results. However, in terms of training efficiency, variations in $\alpha$ lead to noticeable differences. In both `3s_vs_5z` and `6h_vs_8z`, larger values of $\alpha$ correspond to faster learning progress, with the

setting $\alpha = 1$ (when $\mathcal{L}_{\text{tac}}$ and $\mathcal{L}_{\text{main}}$ are of the same order of magnitude), achieving the most rapid ascent in performance. In contrast, the `corridor` scenario exhibits an inverse trend. The smallest $\alpha$ value (1e-5) results in the most efficient training, while $\alpha = 1$ leads to the slowest improvement. This discrepancy suggests that the optimal scaling of the tacit constraint term may depend on the specific task dynamics: in scenarios requiring more active adjustment of inter-agent coordination, a stronger regularization signal could accelerate policy adaptation.

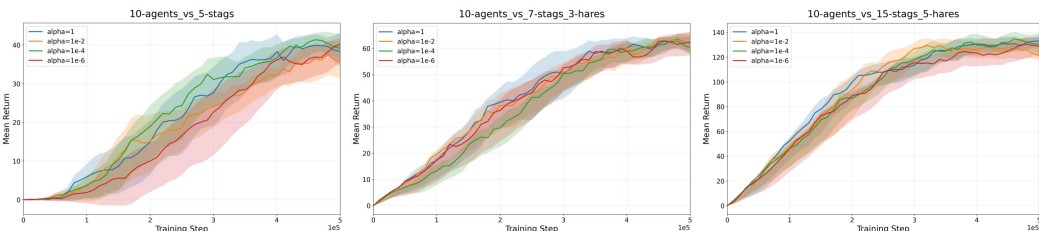

Figure 22: Effect of $\alpha$ parameter on optimization in Predator-Prey tasks.

In the Predator-Prey environment, we evaluate three task configurations: "10 agents vs. 5 stags", "10 agents vs. 7 stags and 3 hares" and "10 agents vs. 15 stags and 5 hares", as shown in Figure 22. For the latter two tasks, the choice of $\alpha$ has limited impact on the overall training performance, with $\alpha = 1$ yielding slightly better results. In contrast, for the more complex "10 agents vs. 5 stags" scenario, the performance varies significantly across different $\alpha$ values. Specifically, both $\alpha = 1$ and $\alpha = 1e-4$ achieve satisfactory performance, while a smaller value such as $\alpha = 1e-6$ leads to noticeably slower convergence. These results suggest that larger $\alpha$ values generally yield better training outcomes across all three settings in Predator-Prey environments. This can be attributed to the fact that the task objective in Predator-Prey is largely aligned with the tacit objective. Therefore, increasing the weight of the tacit constraint term (i.e., using a larger $\alpha$) reinforces beneficial inductive biases without introducing significant conflict with task-specific learning. Conversely, when $\alpha$ is too small, the influence of tacit guidance becomes negligible, diminishing its intended effect, especially in more challenging settings where such guidance is crucial.

## H   ADDITIONAL RESULTS FOR VISUALIZATION ANALYSIS

To further demonstrate PTMC's superior exploration efficiency and its effectiveness in guiding exploration, we collect all global states encountered during 32 evaluation episodes for both PTMC and MAPPO under the same random seed, where PTMC is built on the MAPPO framework. Each state is embedded into a low-dimensional space using t-SNE. Specifically, we evaluate models from the 2M, 4M, 8M, and 10M training steps for both algorithms, treating the 10M-step models as approximations of the optimal policy. To ensure consistent relative positioning across different models, the eight groups of states are jointly embedded into a shared t-SNE space, enabling direct spatial comparison based on preserved relative distances.

**1. Return-colored state embedding:**

Each subplot shows the t-SNE projection of the global states obtained from a specific model. Each scatter point is color-coded based on the normalized cumulative return of its corresponding state, using a gradient from light blue (low return) to pink (high return). The mean return of each subplot is annotated in the lower-left corner for comparison.

**2. State coverage comparison with the optimal model:**

Each subplot presents the projected states of a given model alongside those from its corresponding optimal model (i.e., the model at 10M training steps). Green points represent the states from the current model, while orange points indicate the states from the optimal model. The overlap ratio between the two sets of points is shown in the top-left corner for comparison.

## H.1 VISUALIZATION ANALYSIS ON 3s_vs_5z MAP

For the `3s_vs_5z` map visualization, as shown in Figure 23, the state distribution progressively shifts toward high-return regions as training advances. Compared to optimal-MAPPO, the distribution of optimal-PTMC states is more concentrated in high-return areas, which is also reflected in a higher mean return displayed in each subplot. Notably, PTMC at 8M training steps achieves a mean return comparable to that of optimal-MAPPO. Furthermore, at an early stage (2M steps), PTMC already outperforms MAPPO in terms of both mean return and the extent to which high-return regions are explored. Additionally, at 4M training steps, PTMC not only explores high-return regions more extensively than MAPPO but also surpasses MAPPO's 8M-step performance in terms of mean return. These results demonstrate the superior exploration efficiency of PTMC during coordinated training.

As shown in Figure 24, both PTMC and MAPPO exhibit progressively increased coverage of the corresponding optimal model's state distribution as training advances, indicating convergence toward more effective behavioral patterns. Notably, at 2M training steps, MAPPO demonstrates higher coverage relative to its optimal model than PTMC does. However, by 8M steps, PTMC surpasses MAPPO in overlap ratio, suggesting that PTMC achieves a faster convergence rate toward the optimal policy and thus exhibits superior exploration efficiency.

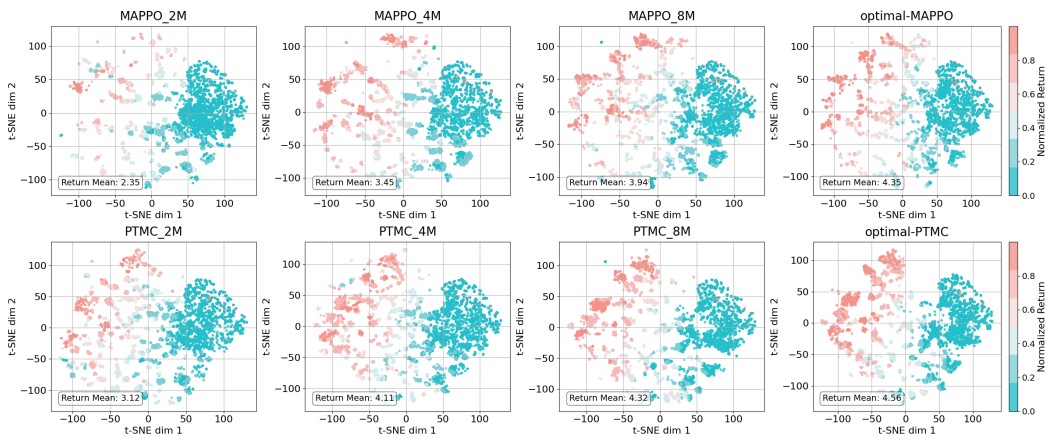

Figure 23: Comparative visualization of return-colored state distributions on 3s_vs_5z map.

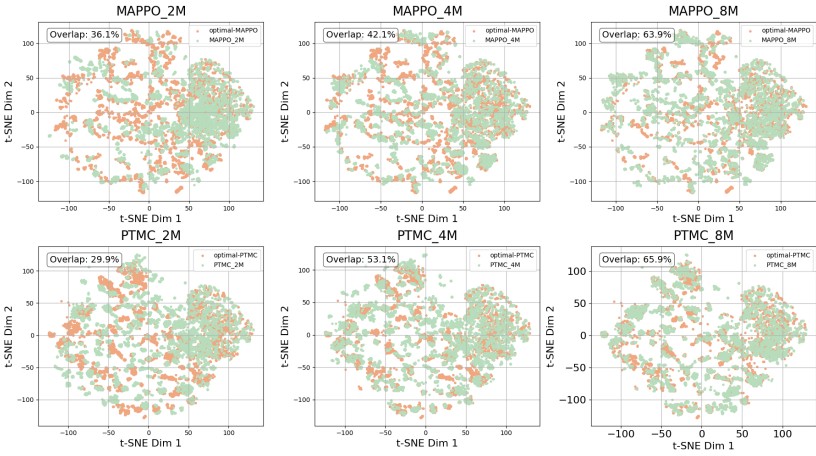

Figure 24: State coverage overlap with the optimal model on 3s_vs_5z map.

## H.2 VISUALIZATION ANALYSIS ON 6H_VS_8Z MAP

For the `6h_vs_8z` map visualization, as shown in Figure 25. Unlike the similar state distributions observed between MAPPO and PTMC on the `3s_vs_5z` map, the high-return regions explored by MAPPO differ significantly from those of PTMC. Overall, PTMC achieves a higher mean return. Notably, at 4M training steps, PTMC already surpasses the mean return of MAPPO at 8M. At 8M training steps, PTMC outperforms the optimal model's mean return achieved by MAPPO. This advantage can be attributed to PTMC's more efficient exploration of high-return states. Furthermore, the distinct explored state regions indicate that PTMC reaches more optimal policy faster than MAPPO.

In the visualization of state coverage on the `6h_vs_8z` map (Figure 26), both MAPPO and PTMC exhibit increasing overlap between the current model and the optimal model as training progresses. However, the overlap ratio of MAPPO at 4M training steps is unexpectedly higher than that at 8M. Given that the mean return of MAPPO at 4M is lower than at 8M (Figure 25), this suggests that the higher overlap at 4M is primarily due to convergence in low-return regions rather than effective exploration of high-return states. Additionally, at 2M training steps, PTMC shows a significantly higher overlap ratio than MAPPO, which can be attributed to the effective initialization of PTMC, further supporting the strength of its algorithmic design.

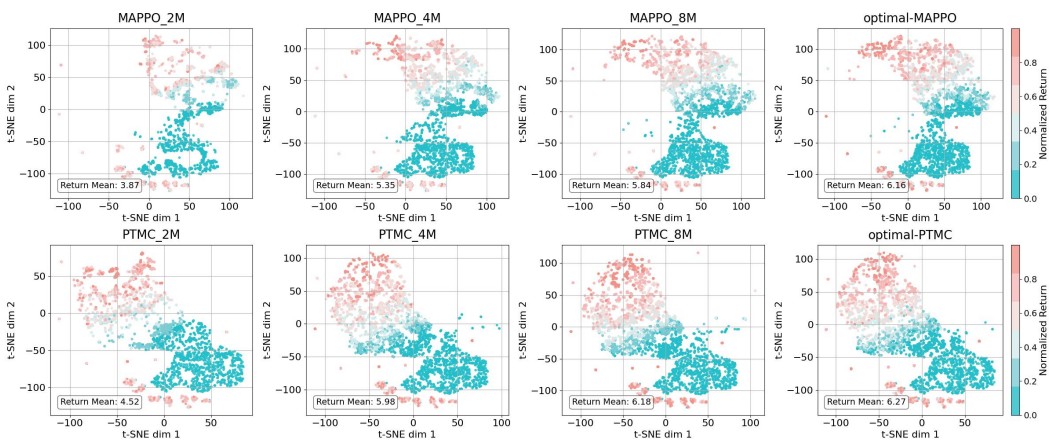

Figure 25: Comparative visualization of return-colored state distributions on 6h_vs_8z map.

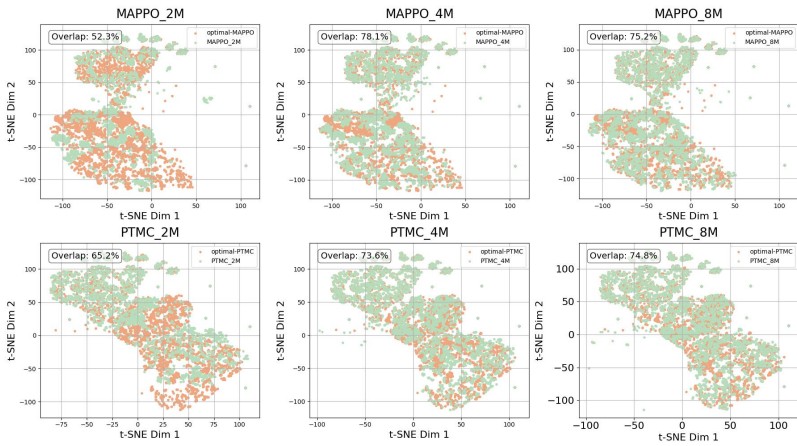

Figure 26: State coverage overlap with the optimal model on 6h_vs_8z map.

### H.3 VISUALIZATION ANALYSIS ON `CORRIDOR` MAP

For the comparative visualization of return-colored state distributions on corridor map, as shown in Figure 27. MAPPO achieves a relatively high return mean at 2M training steps, but fails to maintain this advantage in later stages, indicating limited policy stability and an inability to consolidate early gains. Although both MAPPO and PTMC exhibit increasing return means over time, their learning efficiency differs significantly. Between 4M and 8M steps, MAPPO's return mean improves by only 0.07, whereas PTMC achieves a 0.67 increase, demonstrating PTMC's superior efficiency in discovering high-return states and sustaining policy refinement.

In the visualization of state coverage on `corridor` map (Figure 28), PTMC exhibits an increasing overlap between the current and optimal model as training progresses. In contrast, MAPPO shows a relatively high overlap ratio with the optimal model at 2M training steps. Consistent with its performance in Figure 27, this early overlap likely stems from high-return state exploration but is not maintained in later stages. Moreover, the overall lower overlap ratio values and the distribution of scatter points suggest a wider range of strategy choices on `corridor` map. This is attributed to the scenario's design, which involves 6 ally units and 24 enemies, resulting in a large joint action and policy space. Under such complexity, PTMC demonstrates consistent improvements in return mean, overlap ratio, and the overall trend of state distribution. These metrics progressively align with the optimal model, highlighting PTMC's stability and efficiency.

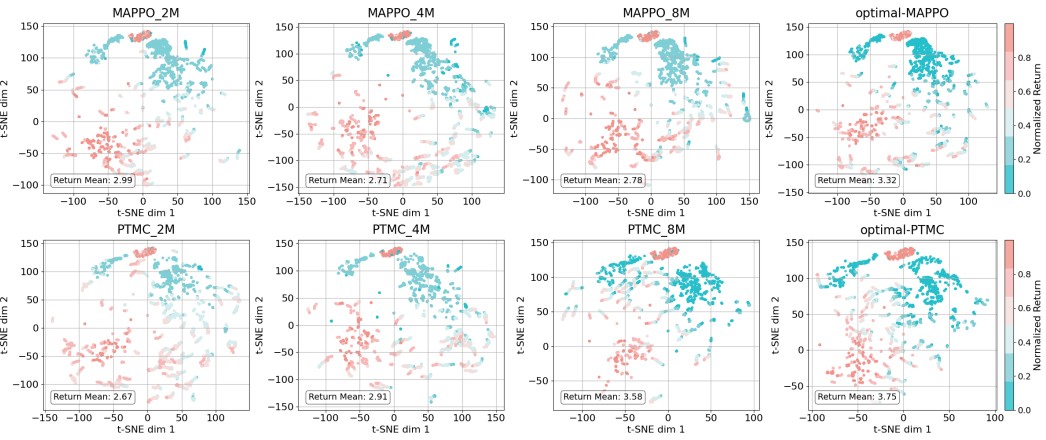

Figure 27: Comparative visualization of return-colored state distributions on corridor map.

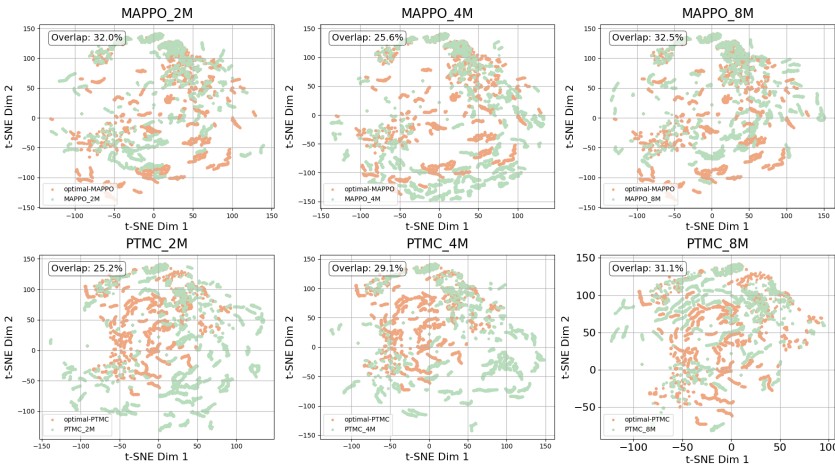

Figure 28: State coverage overlap with the optimal model on corridor map.

