# OpenReview forum: "Leveraging Pre-Trained Tacit Model for Efficient Multi-Agent Coordination"
_ICLR.cc/2026/Conference — ICLR 2026 Conference Withdrawn Submission_

### Official Review · Reviewer_xbJb · 2025-10-15

**Soundness:** 2
**Presentation:** 2
**Contribution:** 2
**Rating:** 2
**Confidence:** 3

**Summary:**

This paper proposes PTMC, a two-phase framework designed to improve exploration and coordination in multi-agent reinforcement learning (MARL). In the first phase, agents undergo tacit pre-training guided by a tacit reward that encodes general cooperative priors without relying on task-specific shaping rewards. This produces a transferable “tacit model” that captures implicit coordination skills. In the second phase, coordinated training initializes from this model and introduces a tacit constraint term to retain useful pre-trained behaviors while adapting to environment rewards. Experiments show that PTMC can enhance training efficiency, coordination quality, and stability compared to some state-of-the-art MARL baselines.

**Strengths:**

The paper presents an innovative use of a pre-trained tacit model to integrate general prior knowledge for improving exploration and coordination in multi-agent reinforcement learning. By introducing tacit consensus and a corresponding tacit reward, it fosters the learning of cooperative behaviors without handcrafted rewards. The two-phase PTMC framework shows good scalability and effectiveness, achieving superior results on StarCraft II and Predator–Prey tasks.

**Weaknesses:**

While the proposed PTMC framework is conceptually novel, its claim of improving exploration efficiency through tacit pre-training lacks solid theoretical justification. The paper does not provide a formal analysis explaining why or under what conditions the tacit reward would lead to more efficient exploration, relying instead on empirical evidence.

Moreover, the effectiveness of tacit training critically depends on how the “advantageous configuration” and corresponding “configuration distance” are defined. These definitions appear to require significant domain knowledge and manual design effort, which undermines the generality and practicality of the approach. In complex or poorly structured environments, such configurations may not be easily identifiable, and subjective or suboptimal definitions could even hinder learning rather than help. Consequently, while the framework demonstrates empirical improvements, its theoretical soundness and scalability to diverse domains remain questionable.

While the authors claim that alternative methods for deriving tacit rewards could be integrated into PTMC, they do not provide concrete examples or clear guidelines on what such methods might be, how they would differ from the current formulation, or how the framework would accommodate them. As a result, it is difficult to assess whether PTMC represents a genuinely new learning paradigm or simply a heuristic way of introducing additional reward signals. The notion of “tacit consensus” is conceptually appealing but remains abstract, with limited formal grounding or demonstrable generality.

The rationale for applying information masking during the tacit pre-training phase is not sufficiently justified. The authors argue that masking opponent-related information encourages agents to learn opponent-agnostic coordination skills, but it remains unclear why this design choice is necessary or optimal for promoting tacit behavior. In fact, removing information may hinder learning by depriving agents of useful context about their environment and teammates, potentially reducing the richness of the pre-trained model. Moreover, if the goal is to capture general cooperative priors, it is not obvious why agents must be trained independently rather than in a loosely coordinated or partially shared setting. Without ablation or theoretical analysis demonstrating that masking indeed leads to better generalization or coordination, this component appears heuristic and insufficiently motivated.

The definition of the *tacit metric* $M_{\text{tac}}$ is problematic in both sensitivity and generality. Using the Heaviside step function effectively binarizes tacit rewards, so even a tiny positive reward contributes equally to $M_{\text{tac}}$ as a large one, failing to capture the actual strength of tacit behavior. Moreover, the reliance on a predefined threshold $M_{\text{tac}}^*$ makes the metric highly task-dependent and may require substantial fine-tuning across different environments. This weakens the claimed efficiency and generality of the tacit pre-training process.

Another key concern is that the proposed algorithm introduces several additional hyperparameters, such as $\epsilon$ and $\alpha_{\text{tac}}$, without a clear discussion of their sensitivity or tuning complexity. It is unclear how difficult it is to calibrate these parameters for each environment, and whether the reported performance improvements stem primarily from extensive hyperparameter fine-tuning rather than from the intrinsic merits of the proposed framework. A systematic sensitivity analysis or justification of default settings would be necessary to support the claimed robustness of the method.

In line with this, the definition of the gating function $G$ in Equation (10) implicitly assumes that the environment reward can take both positive and negative values. This assumption is highly problem-specific and may not hold in many practical settings where rewards are strictly non-negative or sparse. Consequently, the general applicability of the proposed method becomes questionable, as the gating mechanism and its activation logic might fail or require redesign in such environments.

The experimental evaluation is limited, relying only on SMAC/SMACv2 and Predator–Prey benchmarks, which both focus on cooperative, discrete-action tasks with similar dynamics. This narrow scope makes it difficult to assess the generality and robustness of the proposed method across diverse multi-agent settings. Including additional benchmarks would provide stronger evidence of the method’s applicability and scalability.

While the proposed method shows consistent improvements over baselines, the actual performance gains across many tasks appear modest. The increases in win rate and training efficiency are relatively small, suggesting that the benefits of the two-phase framework may not justify its added complexity. Overall, the improvements seem incremental rather than substantial, limiting the claimed impact of the approach.

**Questions:**

1. Why should the tacit reward, defined through heuristic configurations, theoretically lead to better exploration?

2. How sensitive is PTMC to manually chosen configurations and hyperparameters across different benchmarks?

3. Do the limited benchmarks and modest gains truly demonstrate the method’s generality and advantage?

---

### Official Review · Reviewer_dGev · 2025-10-26

**Soundness:** 2
**Presentation:** 3
**Contribution:** 2
**Rating:** 4
**Confidence:** 4

**Summary:**

This paper proposes a novel two-phase MARL framework consisting of a pre-training phase and a coordinated training phase. In the pre-training phase, the authors construct tacit rewards to inject prior knowledge into policy learning while masking opponents’ information. In the coordinated training phase, they incorporate a tacit constraint term into the optimization objective, initialized with the pre-trained policy. The approach is evaluated on SMAC, SMACv2, and Predator–Prey environments.

**Strengths:**

1. The idea of building consensus among agents is great, which is helpful for cooperative MARL.

2. The paper writing is easy to follow.

**Weaknesses:**

1. Practical applicability and generalization: I am concerned about the practical applicability and generalization of the proposed approach. It requires defining tacit rewards tailored to different environments, which may not scale well across tasks. Moreover, it is unclear how the reliability and accuracy of the constructed tacit rewards can be ensured or validated.

2. Potential-based shaping condition: In the Abstract and Introduction, the authors state that intrinsic rewards violating the potential-based shaping condition can induce policy deviation and hinder optimal policy learning. Does the tacit reward introduced in this paper satisfy a potential-based (PBRS) condition, or can it be expressed as a potential difference? Some theoretical analysis or sufficient conditions would be valuable here.

3. Masking opponents’ information: Is it appropriate to mask opponents’ information? The motivation given is to make the tacit policy opponent-agnostic. However, consensus among cooperative agents can still depend on the behavior or positions of opponents. For example, in the toy scenario, if the two apples are far apart, the consensus to move together should arguably change, as agents should search independently. In more complex scenarios where apples can be sometimes close and sometimes far (analogous to moving opponents in SMAC), masking opponents’ information may make it difficult to form robust consensus. Please justify this design choice more rigorously, and discuss the impact on learning stability and performance variability. Moreover, it also lacks an ablation study about masking opponents' information.

4. Construction of Eq. (2): Equation (2) needs more explanation. Why is the tacit reward constructed in this specific form? Please provide the design rationale, ablation or sensitivity analysis, and, if possible, connections to known shaping or decomposition principles.

5. Notation of state s: The notation for the state s is inconsistent: sometimes it is indexed by agent i and sometimes it is not. Under a Dec-POMDP, the environment state is global and not agent-specific; agent-specific quantities are observations or belief states. Please standardize the notation and clearly distinguish between global state s, local observation o_i, and any agent-specific latent representations.

6. Gating function in Eq. (10): In Eq. (10), the gating function G depends on whether the environmental reward is negative or non-negative. In environments like SMAC, the environment reward is typically non-negative (≥ 0). How is the gate activated in such cases?

7. Comparisons to individual-reward methods: The tacit reward can be viewed as a form of individual reward shaping. The paper should include comparisons with individual reward–based methods such as IRAT, MASER, and LAIES.

**Questions:**

Please refer to Weaknesses section.

---

### Official Review · Reviewer_qarp · 2025-10-29

**Soundness:** 2
**Presentation:** 3
**Contribution:** 2
**Rating:** 4
**Confidence:** 5

**Summary:**

This paper introduces PTMC, a two-phase MARL framework designed to improve exploration efficiency by incorporating general prior knowledge. In the first phase, the agents are decentrally pre-trained useing tacit reward with 'tacit consensus' among agents while masking opponent information. In the second phase, coordinated training initalizes policies from the pretrained tacit model and introduces tacit constraint to selectively retain beneficial tacit behaviours while adapting to task-specific goals. The framework is evaluated on SMAC, SMACv2, and Predator-Prey environment and results show outerperformance compared to other baselines.

**Strengths:**

1. Originality: This paper introduces an interesting concept of 'tacit consensus' as methods to encode general prior knowledge in MARL. By formalizing this notion and translating it into tacit reward for decentralized pre-training, the author connect ideas from human teamwork an coordiantion to computational RL.
2. Quality: The proposed PTMC is well-motivated and technically sound. The design of both the tacit reward function and the tacit constraint term is clearly described with clear mathematical formulations. The experiments cover multiple benchmarks, ablation studies and visulization analyses.
3. Clearity: This paper is generally well-written and logically structured. The formal definations of tacit consensus and rewards are clearly explained.
4. Significance: The proposed framework addresses a meaningful challenge in MARL, efficient exploration and coordination under large policy spaces. By proposing a reusable tacit pre-training mechanism that can transfer coordination priors across tasks, the work has potential significance for developing more sample-efficient and scalable MARL systems.

**Weaknesses:**

1. Limited conceptual distinction from existing pre-training framework: While the notion of tacit consensus is interesting in terminology, its functional role resembles representation pre-training or auxiliary reward learning in prior MARL algorithms. This paper could better clarify the differentiates tacit pre-training from skill priors, intrinsic motivation, or behavioral cloning methods.
2. Weak theoretical grounding of tacit constraint term: The introduction of the binary gating function and deviation regularizaion term is intuitive but lacks a theoretical justification or convergence analysis. Providing a theoretical discussion or empirical sensitivity analysis would strengthen this work.
3. Incomplete analysis of transferability and scalability claims: This work claims that the pre-trained tacit model is lighweight and reusable across tasks, however, the experimental section does not demonstrate the cross-task transfer.
4. Clarity and accessibiilty of technical sections: The mathematical formulations in the appendix defining tactic rewards for each envrionment introduces many parameters without sufficient intuition. Condensing and explaining these equation in terms of behavioural implications will help.
5. Besides, as shown in the appendix, the design of tacit consensus requires corresponding adjustments among different envrionments and tasks.
6. The 'randomized initial positions' in SMAC tasks naturally encourages exploration and reduces the performances of other methods. Due to the claim that this paper focuses on the exploration efficiency aspects, the baselines should also contain exploration methods such as MAVEN etc.

**Questions:**

Based on the above discussion, I have some other questions for authors to clarify.
1. The proposed tacit consensus and tacit reward mechanisms resemble prior work in sill pretraining such as [1] and intrinsic reward shaping such as [2,3]. Could the authors clearly articulate what is fundamentally new in PTMC beyond a restructured combination of pre-training and fine-tuning?
2. How does “tacit knowledge” differ from latent behavioral priors or common-sense heuristics used in previous exploration-based methods?
3. The binary gating function and deviation regularization term appear heuristic. Can the authors provide theoretical insights or empirical evidence that this term consistently stabilizes learning rather than acting as a regularizer?
4. Meanwhile, does the constraint affect policy monotonicity or convergence properties in the coordinated training phase?
5. The paper claims that the tacit pre-training is “highly efficient” and “scalable,” but the experiments all use the same task family (SMAC/SMACv2). Could the authors include or describe transfer experiments among opponent strategies, such as SMAC-Hard envrionment?
6. The paper does not discuss how robust PTMC is to the choice of tacit reward coefficients or tacit metric threshold M. Could the authors provide sensitivity analyses for these parameters to ensure the method’s stability?
7. The paper states that the tacit pre-training phase constitutes only a minor fraction of total training time. Could the authors provide concrete ratios (e.g., % of total wall-clock time) and compare them to baseline training times to substantiate this claim?
8. Although during the first phase, the agents are decentrally trained and supervised by the shared tacit reward, the shared tacit reward seems a centralized training process. Does this violate the decentralized training during the first phase?
9. The paper claims that the PRMC can also compatible to value-based algorithm QMIX and AC-based MAPPO. However in Figure 2, there are actor and critic networks training process, which is AC-based demonstration. Then how is PTMC intergrated to value-based QMIX algorithm?
10. In the code provided in the supplementary materials, the QMIX is based on the yaml file from the 'config' folder, which is not provided. What hyper-parameters are used for the experiments. A table of used hyper-parameters will be helpful.

With the concerns all addressed, I would consider raising the score. Thank you.


[1] Karl Pertsch, Youngwoon Lee, and Joseph Lim. Accelerating reinforcement learning with learned skill priors. In Conference on robot learning, pp. 188–204. PMLR, 2021.
[2] Lulu Zheng, Jiarui Chen, Jianhao Wang, Jiamin He, Yujing Hu, Yingfeng Chen, Changjie Fan, Yang Gao, and Chongjie Zhang. Episodic multi-agent reinforcement learning with curiosity-driven exploration. Advances in Neural Information Processing Systems, 34:3757–3769, 2021.
[3] Xinran Li, Zifan Liu, Shibo Chen, and Jun Zhang. Individual contributions as intrinsic exploration scaffolds for multi-agent reinforcement learning. In International Conference on Machine Learning, pp. 28387–28402. PMLR, 2024.

---

### Official Review · Reviewer_LfC7 · 2025-10-31

**Soundness:** 2
**Presentation:** 2
**Contribution:** 2
**Rating:** 2
**Confidence:** 3

**Summary:**

This paper considers a multiagent coordination problem and solves the problem using RL. A key difference compared to past works is that the training is explicitly divided into pre-training and coordinated training, where non-environmental priors are baked into the pre-trained "tacit" model and further be used in the training for coordination. The authors show that the proposed method outperforms its baselines.

**Strengths:**

It is easy to follow the paper. The motivation is clear. In MARL, there are many aspects of the problem might not be easily defined as the final reward yet are important for explorations. The two phase training idea is interesting. Standard techniques are used to avoid certain "forgetting" behavior, e.g., KL regularization and the gating function.

**Weaknesses:**

At the beginning, I thought the "pre-training" is done similarly as in LLM, where a generic task is given, e.g., predicting next token. In MARL, the task might be predict the location of nearby agents. However, in the paper, "pre-training" is more task-related.  In the experiments, the author show that by using the proposed PTMC, the PTMC augmented MARL methods outperforms the base ones. I am not convinced that whether this improvement is due to the extra reward information used in tacit or the PTMC framework itself.

**Questions:**

Have you done any experiments to argument the environment reward with the priors? For example, the final reward = env reward + alpha * tac reward. It would be good to understand the benefits of using pre-training by comparing to a baseline that uses tac reward.

**Details Of Ethics Concerns:**

n.a

---

### Note · Authors · 2025-12-20

I have read and agree with the venue's withdrawal policy on behalf of myself and my co-authors.